# Spatial distribution and controls of snowmelt runoff in a sublimation-dominated environment in the semiarid Andes of Chile

Álvaro Ayala[1], Simone Schauwecker[1], Shelley MacDonell[1, 2]

[1] Centro de Estudios Avanzados en Zonas Áridas (CEAZA), La Serena, 1700000, Chile

[2] Waterways Centre for Freshwater Management, Lincoln University and the University of Canterbury, Christchurch, New Zealand

*Correspondence to*: Álvaro Ayala (alvaro.ayala@ceaza.cl)

**Abstract.** Sublimation is the main ablation component of snow in the upper areas of the semiarid Andes (~26-32°S and ~69-71°W). This region reaches up to 6000 m, is characterized by scarce precipitation, high solar radiation receipt, and

low air humidity, and has been affected since 2010 by a severe drought. In this study, we suggest that most of the snowmelt runoff originates from specific areas with topographic and meteorological features that permit large snow accumulation, limited removal by sublimation and sufficient energy for snowmelt. To test this hypothesis, we quantify the spatial distribution of snowmelt runoff and sublimation in a catchment of the semiarid Andes using a process-based snow model that is forced with field data. Model simulations over a two-year period reproduce point-scale

records of snow depth and SWE and are also in good agreement with an independent SWE reconstruction product as well as satellite snow cover area and indices of snow winter absence and summer persistence. We estimate that 50% of snowmelt runoff is produced by 21-29% of the catchment area, which we define as "snowmelt hotspots". Snowmelt hotspots are located at mid-to-lower elevations of the catchment on wind-sheltered, low-angle slopes. Our findings show that sublimation is not only the main ablation component, but it also plays an important role shaping the spatial

variability of total annual snowmelt. Snowmelt hotspots might be connected with other hydrological features of arid and semiarid mountain regions, such as areas of groundwater recharge, rock glaciers and mountain peatlands. We recommend a more detailed snow and hydrological monitoring of these sites, especially in the current and projected scenarios of scarce precipitation.

## 1 Introduction

Snowmelt is typically the largest runoff contributor in high-elevation mountain regions and their adjacent areas (Mankin et al., 2015; Kraaijenbrink et al., 2021). The annual volume of snowmelt runoff generated from each catchment is the result of the interactions between the topography, the land cover and the physical processes that control snowfall, snow transport, the surface energy balance and the internal changes of the snowpack (Lehning et al., 2008; Mott et al., 2018; Pomeroy et al., 1998). A good understanding of these processes and how to model them is key to quantify water supply availability for several populated regions around the world (Freudiger et al., 2017; Hock et al., 2017). A particularly critical case of snowmelt dependency is that of arid and semiarid high-elevation mountain ranges, such as the semiarid Andes, Central Asia, and Southwestern USA (Huning and AghaKouchak, 2020). The climate of these mountain ranges is characterized by low temperatures and little precipitation, and pose characteristic challenges to estimate snowmelt runoff, such as episodic precipitation events (Schauwecker et al., 2022), shallow snowpacks (Zhang and Ishikawa, 2008) and high sublimation rates (Stigter et al., 2018).

Sublimation from snow cover is the direct transition of water from solid to vapor state and it occurs as surface sublimation (e.g. Hood et al., 1999), or as blowing (or drifting) snow sublimation (e.g. Groot Zwaaftink et al., 2013). Both surface and blowing snow sublimation reduce the mass of the snow cover and can significantly affect the water balance of various regions around the world, such as the Canadian Prairies (Pomeroy and Li, 2000), Antarctica (Palm et al., 2017), and the semiarid Andes (Réveillet et al., 2020; Gascoin et al., 2013). Additionally, turbulent latent heat fluxes associated with the solid to vapor transition use energy from the snowpack, lowering its temperature and decreasing the energy available for melting. This process can have impacts on glacier mass balance (Ayala et al., 2017a; Stigter et al., 2018) and the performance of temperature-index melt models (Ayala et al., 2017b; Litt et al., 2019). Blowing snow sublimation depends on the amount of transported snow and sublimation rates and is usually important in the mass balance of open and windswept environments (Pomeroy and Li, 2000), and wind-exposed mountain ridges (Strasser et al., 2008).

The interplay of physical processes controlling the evolution of the seasonal snowpack results in a large spatial variability of snowmelt and surface sublimation. While the spatial distribution of snow accumulation is controlled by preferential deposition, wind redistribution and gravitational transport (Mott et al., 2010; Freudiger et al., 2017), the energy balance is controlled by solar radiation and snow-atmosphere interactions that evolve during the ablation season (Mott et al., 2018; Pomeroy et al., 2003) and generate large differences in snowmelt rates across a certain domain (Pohl et al., 2006; DeBeer and Pomeroy, 2017). By the end of the ablation season, the snow cover often reaches a patchy state in which the advection of sensible and latent heat from snow-free areas plays a key role for snow and ice ablation (Mott et al., 2020; van der Valk et al., 2022). Solving the energy balance of snow cover over complex and steep terrain is a very complex task, as it depends on the availability of distributed meteorological forcing data and adequate parameterizations of the physical processes controlling radiation, turbulent fluxes, snow microstructure and snow metamorphism (Vionnet et al., 2012; Lehning et al., 2002). Despite this complexity, in hydrological models the temporal and spatial variability of the energy balance is usually averaged over elevation

bands or other hydrological units, but with negative effects on the representation of snow ablation and runoff at the basin scale (Dornes et al., 2008; Luce et al., 1998). Moreover, in the case of dry mountain environments, turbulent latent heat fluxes and
sublimation can represent a large part of the snow cover energy and mass balance (Jackson and Prowse, 2009; Zhang and Ishikawa, 2008), suggesting a particularly large spatial variability of snowmelt and runoff generation. However, snow sublimation is frequently neglected in hydrological models (e.g. Ragettli et al., 2014; Seibert et al., 2017).

The semiarid Andes is a good example of a snowmelt-dependent region where snow sublimation can significantly reduce the mass of the winter snow cover and complicate the calculation of snowmelt runoff. Several studies have estimated snow mass
and energy balances (Gascoin et al., 2013; Réveillet et al., 2020; Voordendag et al., 2021), and have provided useful insights into processes occurring in this region, as well as into the challenges of estimating distributed snowmelt and sublimation in this environment. However, these studies have largely focused on either model or input uncertainties (Réveillet et al., 2020; Voordendag et al., 2021), or snow redistribution (Gascoin et al., 2013), rather than melt processes and hydrological implications. From a geostatistical perspective, Mendoza et al. (2020) analyzed the spatial properties of a set of Lidar snow
depth measurements across several catchments of central Chile and found a strong relation between snow depth and local topographic and land cover properties. Given that snow represents 85% of streamflow variability in semiarid catchments (Masiokas et al., 2006) and is a useful predictor of streamflow (Sproles et al., 2016), it is of vital importance to connect distributed snow mass and energy balance processes to runoff generation in their full complexity and spatial variability.

In this work, we hypothesize that the meteorological and topographical conditions of the semiarid Andes result in large areas
where snow surface sublimation losses dominate over snowmelt, thus delimiting relatively small areas from where most of the snowmelt runoff is generated. We argue that the typically large spatial variability of snowmelt in mountain terrain is further enlarged in dry environments by large sublimation rates that almost fully deplete the snow mass available for melt at wind-exposed sites. We refer to the areas producing most of the snowmelt runoff as "snowmelt hotspots" and define them as the minimum area in a catchment where 50% of the snowmelt runoff is generated. To test this hypothesis, we calculate spatially
distributed amounts of snowmelt and snow sublimation using a process-based snow evolution model in a 79 km$^2$ catchment of the semiarid Andes of Chile over a two-year period (April 2019 to March 2021). The model is forced with in-situ meteorological data, and verified against point observations of snow variables and a set of independent satellite-derived products. Our main objectives are to i) quantify the snow mass balance components, ii) determine the spatial distribution of snowmelt runoff and proof the existence of snowmelt hotspots, and iii) identify the main characteristics of the snowmelt
hotspots and discuss their possible connection with other hydrological components. As our selected study catchment contains glacierized areas, we also include estimates of ice melt and glacier runoff. We expect that our results will provide new insights for studies focusing on snowmelt runoff generation in dry mountain environments and its connection with other components and processes, such as glaciers, permafrost, groundwater recharge and the spatial distribution of vegetation.

## 2 Study area

The semiarid Andes of Chile extend from approximately 26°S to 32°S (~69-71°W) reaching elevations of more than 6000 m a.s.l. Its climate is defined as cold semi-arid with dry summers in the low-lying areas up to approximately 3000 m a.s.l. and as tundra climate in the upper areas (Sarricolea et al., 2017). While winters (June-August) are cold with occasional precipitation events, summers (December-February) are hot and dry with low cloudiness and intense solar radiation (Favier et al., 2009). Precipitation has a large interannual variability mostly associated with El Niño Southern Oscillation (ENSO) (Arias et al.,

2021; Montecinos and Aceituno, 2003). Annual precipitation in the lowlands increases from ca. 100 to 300 mm $a^{-1}$ towards south and is 2-3 times higher at 3000 m a.s.l. (Scaff et al., 2017; Favier et al., 2009). Cloud cover and air moisture are usually very low, which determines very high values of solar radiation and potential evapotranspiration (MacDonell et al., 2013). Although mass losses produced by snow sublimation can reach up to 50 or 80% of the annual snowfall (Réveillet et al., 2020), the hydrological regime of the rivers in this region is mostly nival (Favier et al., 2009). Glacier runoff is an important

contributor during droughts and at the end of summer (Gascoin et al., 2011; Ragettli et al., 2014).

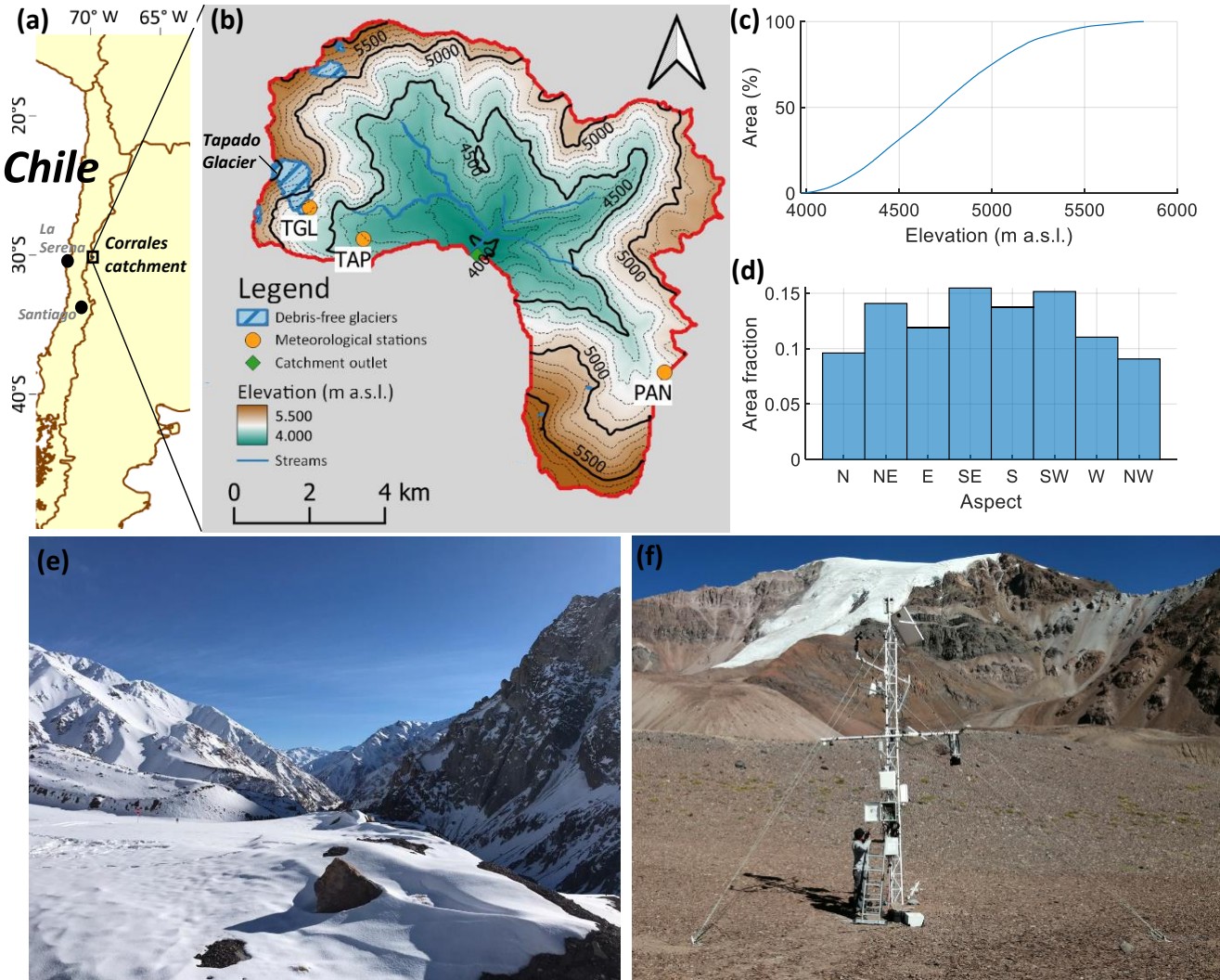

**Figure 1: (a): Location of the Corrales catchment in Chile, (b) Map of the Corrales catchment including Tapado Glacier and the Tapado (TAP), Tapado Glacier (TGL) and Paso Agua Negra (PAN) meteorological stations, (c) Hypsometry of the basin, (d) Slope aspect frequency distribution in the catchment, (e) Winter snow accumulation close to La Laguna reservoir (~3200 m a.s.l.) (Photo: La Laguna Reservoir personnel, 17/06/2020), (f) Tapado AWS with Tapado Glacier in the background (Photo: Cristian Orrego, 22/11/2010).**

This study focuses on the Corrales catchment, a 79 km$^2$ catchment (30.16°S, 69.88°W) located in the upper areas of the semiarid Andes of Chile, approximately 180 km east of La Serena city (Figure 1a-b). The elevation of the catchment ranges from 3800 to 5600 m a.s.l., with ca. 50% of the catchment located above 4700 m a.s.l. (Figure 1c). The orientation of the slopes is relatively uniform across the catchment (Figure 1d). Meteorological records from the nearby La Laguna meteorological station (30.20°S, 70.04°W, 3160 m a.s.l., maintained by the Chilean Water Directorate (DGA) show an annual mean precipitation of 160 mm a$^{-1}$ (with a coefficient of variation of 0.68) and an average temperature of 8.1°C in the period 1965-

2020. Snow varies spatially across the catchment (Figure 1e), and resultant runoff leaving the Corrales catchment is stored in

La Laguna Reservoir (storage capacity of 40 Mm$^3$) from where it is regulated for irrigation and other uses. Above 3000 m a.s.l. vegetation is almost completely absent except for peatlands next to streams (Valois et al., 2020) and shrubs on the mountain slopes (Kalthoff et al., 2006). The Corrales catchment contains Tapado Glacier, which is the largest glacier on the Chilean side of this region (1.67 km$^2$ in 2019) (DGA, 2022). Tapado Glacier has lost $25.2 \pm 4.6\%$ of its area since 1956, with an accelerated negative mass balance since 2000 (Robson et al., 2022), which is in line with other glaciers in the area (Pitte et al., 2022). In

addition, the Corrales catchment contains a few small debris-free glaciers located above 5000 m a.sl. (Figure 1b), and several rock glaciers (not shown) that might play a relevant hydrological role by active layer thawing and by storing and releasing meltwater from snow and ice (Schaffer et al., 2019; Navarro et al., 2023a). Since 2009, the Corrales catchment has been instrumented with meteorological equipment and several glaciological field campaigns have been carried out (e.g., Figure 1f).

## 3 Data

**3.1 Field data**

We use meteorological data from a network of stations in the Corrales basin to force a snow evolution model that is presented in Section 4. The meteorological network consists of three Automatic Weather Stations (AWSs): Tapado (TAP), Tapado Glacier (TGL) and Paso Agua Negra (PAN) (Figure 1 and Table 1). The AWSs are maintained by the Centro de Estudios Avanzados en Zonas Áridas (CEAZA), based in La Serena, Chile, and are visited several times each year between October

and May when the sites are accessible by road. TAP and PAN consist of meteorological towers (Figure 1f, S1a) and transmit near-real time data via satellite communication. TGL is a HOBO U30 station installed on a tripod next to the lower debris-free section of Tapado Glacier that is visited with a lower frequency (2 to 3 times per year) due to logistical limitations (Figure S1b). Data from TGL is downloaded manually.

Hourly precipitation is derived from the cumulative precipitation data recorded at TAP (Table 1). As the cumulative

precipitation record contains noise, we apply a simple method that discriminates actual precipitation from sensor noise (see Section S1). The method is based on the identification of positive changes in the daily precipitation cumulative record that lead to increases in the 5-day moving average of the same series. Additionally, based on a comparison with the SWE sensor and results from previous studies at TAP (Réveillet et al., 2020; Voordendag et al., 2021) and other regions (MacDonald and Pomeroy, 2007), we increase precipitation amounts by 30% due to snow undercatch by the sensor and precipitation

underestimation due to differences between the exact location of the precipitation sensor and the meteorological tower where snow depth and SWE are recorded. Following these corrections, we obtain total precipitation amounts of 309 mm and 330 mm in the hydrological years 2019-2021 and 2020-2021 (from April to March), respectively (Figure 2a). Although annual precipitation was higher in 2020-2021, precipitation in the main snow season (April-September) was higher in 2019 (238 mm compared with 196 mm). Annual precipitation at TAP was 4 to 5 times higher than at La Laguna DGA, where a total of 63

and 82 mm were registered in 2019-2020 and 2020-2021, respectively. Although we do not have a long-term precipitation at

this particular site, we verified that, according to La Laguna DGA precipitation records (Información Oficial Hidrometeorológica y de Calidad de Aguas en Línea, 2023), our study years (2019-2020 and 2020-2021) corresponded to a dry period, with an annual exceedance probability of ca. 80% for both years.

Figure 2 shows a summary of the data collected by the three AWSs during the study period and Table 1 includes additional
details, such as the sensors and their installation heights. Apart from occasional gaps at each station, the largest period of missing data is found at TAP between December 2020 and January 2021. Daily mean air temperature at the elevation of TAP remained above 0°C for several months (November to April), with a maximum near 10°C in January 2020 (Figure 2a), although there was a large data gap in summer 2021. Since the correlation between the three air temperature records is very high (correlation coefficients are higher than 0.97), missing data was filled by establishing linear relationships between the three
stations. Figure 2b shows the variability of snow depth, SWE and surface albedo at TAP. Snow depth reached up to 1 m in winter 2019, but only 0.8 m in winter 2020. In general, snow depth decreases rapidly during the days following each snowfall event. SWE and surface albedo closely followed snow depth variations. We found that the decay of surface albedo was faster in spring 2019 than in 2020. Mean daily solar radiation reached values over 400 W m$^{-2}$ with frequent drops associated with cloudiness, especially during the winter period (Figure 2c). The seasonal variability of incoming longwave radiation was much
lower than that of solar radiation and varies between 180 and 300 W m$^{-2}$ (Figure 2c). Daily mean relative humidity at TAP remained below 50% for most of the time, increasing above this value only a few days during the study period (Figure 2d). Relative humidity at the other stations was similar (not shown). Air pressure varied between 550 and 580 hPa with large and relatively constant values in summer and low and oscillating values in winter (Figure 2d). Wind speed at PAN was 2.6 times larger than at TAP and TGL, with a large predominance of westerly winds, indicating that the latter are located at wind-
sheltered locations (Figure 2e and 2f).

In addition to the meteorological data, we use the inflow to the La Laguna reservoir as a reference for streamflow variations in response to snow and ice melt. These data were derived from the water balance of the reservoir and were provided by the organization in charge of the operation of the reservoir (Junta de Vigilancia del río Elqui y sus afluentes). Finally, we use a dataset of surface elevation changes of Tapado Glacier to validate ice melt estimates produced by the snow evolution model
(Section 4). These observations correspond to ablation stake readings collected in the summer periods 2019-2020 and 2020-2021, and snow density measurements at the end of the corresponding spring periods (Figure S2 and Table S1).

**Table 1: AWSs in the Corrales catchment**

| Station | Name | East (WGS84 UTM 19S) | South (WGS84 UTM 19S) | Elevation (m a.s.l.) | Monitored variables (*) | Instrument | Sensor height (m) |
|---------|------|------|-------|-----------|-----------|------------|------------|
| TAP | Tapado | 412546 | 6663325 | 4306 | T, RH | Vaisala HMP45C | 4.0 |

| | | | | | WS, WD | RM Young 5103 Wind Monitor | 5.0 |
|---|---|---|---|---|---|---|---|
| | | | | | P | Geonor T-200B 1000 mm | 1.5 |
| | | | | | Sin, Sout, Lin | Kipp and Zonen CNR4 | 3.5 |
| | | | | | SD | Luft SHM31 | 3.5 |
| | | | | | SWE | Campbell CS725 | 3.0 |
| TGL | Tapado Glacier | 411121 | 6664158 | 4727 | T, RH | HOBO S-THB-M002 | 2.0 |
| | | | | | WS | HOBO S-WSA-M003 | 2.0 |
| | | | | | WD | HOBO S-WDA-M003 | 2.0 |
| | | | | | Sin, Sout | HOBO S-LIB M003 | 2.0 |
| PAN | Paso Agua Negra | 420534 | 6659795 | 4774 | T, RH | Vaisala HMP155 | 3.0 |
| | | | | | WS, WD | Campbell CSAT3 | 5.0 |
| | | | | | Pa | Vaisala PTB110 | 2.0 |

(*): T: Air temperature, RH: Relative humidity, WS: Wind speed, WD: Wind direction, P: Precipitation, Sin: Incoming solar radiation, Sout: Reflected solar radiation, Lin: Incoming longwave radiation, SD: Snow depth, SWE: Snow water equivalent, Pa: Air pressure.

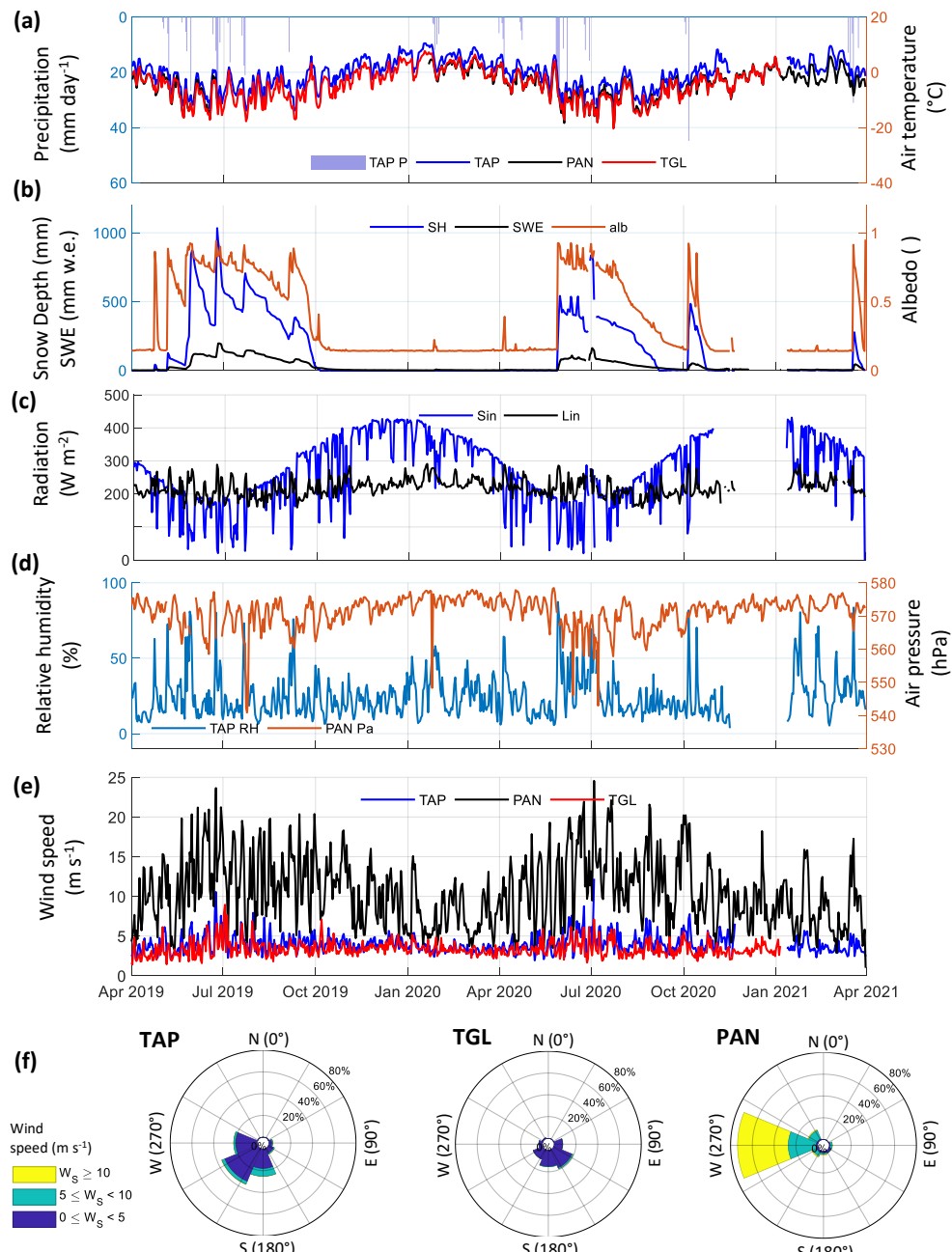

**Figure 2: Meteorological time series recorded at Tapado (TAP), Tapado Glacier (TGL) and Paso Agua Negra (PAN) during the study period. If not indicated, the meteorological variables were recorded at TAP. (a) Precipitation and air temperature, (b) Snow depth, SWE, and albedo, (c) Incoming shortwave and longwave radiation, (d) Relative humidity and air pressure, (e) Wind speed, (f) Wind roses.**

## 3.2 Snow products

We use three spatially distributed datasets that describe the interannual and seasonal variations of the snow cover to evaluate the results of the snow evolution model (Section 4). The first dataset corresponds to satellite-based maps of snow absence and persistence indices (Wayand et al., 2018) in the period 2019-2021, the second to SWE daily maps that were reconstructed using a data assimilation scheme for the subtropical Andes in the period 1985-2015 (Cortés and Margulis, 2017). Finally, we use Sentinel-2 images to derive snow cover area (SCA).

### 3.2.1 Maps of snow absence (SA) and snow persistence (SP) indices

We use the snow indices defined by Wayand et al. (2018) to characterize the spatial patterns of snow cover in the study area. The snow absence (SA) index is defined as the fraction of time in which snow is absent during the accumulation period, whereas the snow persistence (SP) index is defined as the fraction of time in which snow is present during the melt period. In their study, Wayand et al. (2018) developed a method to estimate SA and SP from optical satellite images obtained by Sentinel-2 and Landsat-8 missions. The method was implemented in the Google Earth Engine platform, which is freely available. In our study, we adapt the definition of the snow accumulation and melt periods to the Southern Hemisphere (accumulation: April-September and melt: October-March) and apply the method to the Corrales catchment. For simplicity we refer to the indices as snow winter absence and snow summer persistence. A summary of the utilized images is given in Table 2 (see Table S2 for the full list of images).

**Table 2: Summary of satellite images used for the calculation of the snow indices (SA and SP)**

| Snow index | Months | Product | Period | Number of images |
|---|---|---|---|---|
| Snow winter absence | April to September | Landsat-8 | 13-04-2019 to 22-09-2020 | 18 |
| | | Sentinel-2 | 03-06-2019 to 10-09-2020 | 15 |
| Snow summer persistence | October to March | Landsat-8 | 06-10-2019 to 17-03-2021 | 21 |
| | | Sentinel-2 | 15-11-2019 to 14-03-2021 | 23 |

### 3.2.2 Maps of reconstructed SWE

We analyze the spatial variability of snow accumulation in the study area using the SWE reconstruction developed by Cortés and Margulis (2017) (hereafter C&M2017). This dataset consists of daily maps of SWE with a spatial resolution of 180 m over the extratropical Andes (27-37°S) in the period 1985-2015. The maps were calculated using a data assimilation scheme that combines results from a Land Surface Model and Landsat optical images. The data assimilation scheme explicitly addresses the uncertainty of input variables by weighting simulation ensembles to fit observed fractional snow cover area (fSCA) from the satellite images. Reconstructed SWE maps compare well with manual records along the Andes (Cortés and Margulis, 2017; Cortés et al., 2016) and melt season runoff records (Álvarez-Garretón et al., 2018). Later results have shown that these SWE

maps are suitable for other analyses, such as the impact of atmospheric rivers on snow accumulation (Saavedra et al., 2020)
and the validation of hydrological models (Ayala et al., 2020). In our study, we summarize the information provided by the SWE reconstruction product by calculating the interannual median and coefficient of variation of SWE at the time of annual maximum accumulation in the catchment (hereafter SWEmax and SWEmaxCV). The equation for the coefficient of variation at each grid cell is:

$$CV = \frac{\mu}{\sigma}, \tag{1}$$

where μ is the mean and σ is the standard deviation of peak annual SWE during the period 1985-2015.

### 3.3.3 Sentinel-2 NDSI

We use the Normalized Difference Snow Index (NDSI) derived by the Sentinel-2 Level 2A processing (Level-2A Algorithm Overview, 2023) to calculate SCA over the study catchment and compare it with our model simulations (see Section 4). We manually selected a total of 103 cloud-free NDSI maps in the period April 2019-March 2021 using the EO Browser
(https://apps.sentinel-hub.com/eo-browser/).

## 4 SnowModel

### 4.1 Model description

We use SnowModel to calculate snowmelt, sublimation, and other snow-related variables in the Corrales catchment.
SnowModel is a numerical model for the spatially-distributed simulation of snow evolution with an explicit consideration of the main physical processes that shape the seasonal snowpack (Liston and Elder, 2006a). The model consists of four modules: a module that interpolates meteorological variables recorded at specific locations to a two-dimensional grid using specially developed algorithms (MicroMet) (Liston and Elder, 2006b); a module that solves the energy balance and yields surface sublimation and snowmelt (EnBal) (Liston, 1995); a module that solves the internal changes of the snowpack, such as
refreezing, densification and metamorphism (SnowPack) (Liston and Hall, 1995); and a module that solves the snow transport and blowing snow sublimation due to saltation and suspension of the snow (SnowTran-3D) (Liston et al., 1998). The model does not include a representation of snow gravitational transport (avalanches). SnowModel has been successfully tested in several snow environments around the world (Liston et al., 2007; Mernild et al., 2016), including the semiarid Andes (Gascoin et al., 2013; Réveillet et al., 2020; Voordendag et al., 2021). Readers are referred to other studies for a full description of the
model (Liston and Elder, 2006a) and its updates (Mernild et al., 2018; Merkouriadi et al., 2021).
In SnowModel, snowmelt either refreezes in lower layers or drains as runoff when the snowpack has already reached a ripe state. In our study, we analyze results from snowmelt runoff (snowmelt leaving the snowpack) and snow surface sublimation (extracted from the surface turbulent latent heat flux). Rain on snow contributes to a snow density increase until reaching a

maximum density of 550 kg m$^{-3}$. Beyond that limit, rain on snow is added to snowmelt runoff. We define the sublimation

fraction (SublFrac) as shown in Eq. (2):

$$SublFrac = \frac{Snow\ surface\ sublimation}{Snow\ surface\ sublimation + Snowmelt\ runoff} \qquad (2)$$

Glaciers can be included in SnowModel as a type of surface and ice is melted once the snow has completely disappeared from the surface. Debris-covered and rock glaciers are not included in the model.

## 4.2 Setup

We run SnowModel in the Corrales basin using a 3 h timestep in the period April 2019-March 2021. The domain of the model runs is a rectangle that contains the Corrales basin with an additional buffer of 500 m in all four main directions (N-S-E-W). We use a DEM of a spatial resolution of 50 m that is bilinearly resampled from a ~30 m resolution Digital Elevation Model (DEM) produced by NASADEM (NASA JPL, 2020). The land cover of the model domain consists entirely of bare soil except for the debris-free glaciers shown in Figure 1b. The DEMs and related variables are analyzed using Topotoolbox functions in

MATLAB (Schwanghart and Scherler, 2014).

**Table 3: Parameters values used in the SnowModel simulations**

| Module | Parameter | Value or source | Units |
|---|---|---|---|
| General | Spatial resolution | 50 | m |
| | Time step | 3 | h |
| | Number of grid cells (East, North) | 270, 272 | |
| Meteorological inputs | Precipitation | TAP | mm |
| | Air temperature | TAP, TGL, PAN | °C |
| | Relative humidity | TAP, TGL, PAN | % |
| | Wind speed and direction | TAP, TGL, PAN | m s$^{-1}$ and ° |
| | Solar radiation | TAP | W m$^{-2}$ |
| | Incoming longwave radiation | TAP | W m$^{-2}$ |
| | Air pressure | PAN | Pa |
| MicroMet | Precipitation factor | 0.7, 1.0, 1.3 | |
| | Curvature length scale | 500 (default) | m |
| | Slope weight for wind distribution | 0.25, 0.58 (default), 0.75 | |
| | Curvature weight for wind distribution | 0.25, 0.42 (default), 0.75 | |
| | Monthly mean air temperature lapse rates | From January to December: 8.1,8.1,7.7,6.8,5.5,4.7, 4.4,5.9,7.1,7.8,8.1,8.2 (default) | °C km$^{-1}$ |

| | | | |
|---|---|---|---|
| | Precipitation and wind speed lapse rate | 0 | |
| | Rain-snow air temperature threshold | 2 (default) | °C |
| Enbal | Albedo decay melt conditions | 0.024 | $s^{-1}$ |
| | Albedo decay cold conditions | 0.008 | $s^{-1}$ |
| | Ice albedo | 0.3 | |
| | Albedo fresh snow | 0.9 | |
| | Soil albedo | 0.14 | |
| | Precipitation threshold for albedo reset | 0.006 (default) | m |
| | Aerodynamic surface roughness length for snow ($z_0$) | 0.001 (default), 0.005, 0.010 | m |
| SnowPack | Maximum number of snow layers | 6 | |
| SnowTran-3D | Threshold surface shear velocity | 0.25 (default) | $ms^{-1}$ |

We run the model using the parameters shown in Table 3 and without performing any calibration except for the albedo decay rates, which were manually set to fit the albedo changes observed at TAP (Figure S4). The model is forced using the
meteorological field data described in Section 3.1. The wind speed records were adjusted from the corresponding sensor height to a height of 2 m using a logarithmic wind profile and an aerodynamic surface roughness length for snow of 5 mm. Although there is an annual precipitation lapse rate from the lowlands of the Coquimbo Region up to La Laguna DGA station (3160 m a.s.l.), we used a value of zero because we do not have enough data to support a precipitation lapse rate above that elevation, particularly within the relatively small area of the Corrales catchment. In general, snow distribution at high-elevation
catchments is governed mostly by wind transport (e.g. Lehning et al., 2011).

### 4.3 Ensemble runs

We produce ensemble runs considering three different values for three selected variables: a precipitation factor, the aerodynamic surface roughness length for snow (hereafter $z_0$) and two wind distribution weights based on the slope and curvature of the terrain in the MicroMet module (Table 3). Previous snow simulations in this area have resulted to be most
sensitive to these variables (Réveillet et al., 2020; Voordendag et al., 2021). We selected precipitation factors that can be interpreted as an uncertainty range for snow undercatch. The aerodynamic surface roughness length is a key parameter controlling the mass and energy exchanges between the surface and the atmosphere and has been defined as the height above a surface at which the extrapolated horizontal wind-speed profile reaches zero (Brock et al., 2006). The selected values for $z_0$ vary within typical ranges for snow and ice surfaces (Brock et al., 2006; Fitzpatrick et al., 2019). The slope and curvature
distribution weights increase wind speed in the presence of windward and convex slopes and decrease it in the case of leeward and convex ones (Liston et al., 1998). According to the MicroMet module, the slope and curvature wind distribution weights

should sum to one. The selected values for these variables were chosen to explore the sensitivity of snow ablation to these parameters in the semiarid Andes. We test all the different combinations of these values, obtaining a total of 27 simulations that are used to assess model sensitivity and uncertainty.

## 5 Results

### 5.1 Analysis of snow products

The analysis of snow winter absence (SA), snow summer persistence (SP) and the interannual median of peak annual SWE (SWEmax) shows that the north-west section of the catchment present larger values of snow accumulation and persistence than the areas to the east. While SA and SWEmax values on the north-west of the catchment are lower than 0.6 (Figure 3a) and larger than 400 mm $a^{-1}$ (Figure 3g), respectively, the areas to the east present the opposite behavior (SA>0.8 and SWEmax<200 mm $a^{-1}$). We further connect these patterns to the underlying topography by means of polar plots (Figure 3b-e-h). These plots show that while the lowest values of SA (SA<0.6) are found at sites higher than 4000 m a.s.l. with a SW, S or SE aspect, the highest values (SA>0.8) are found at sites with a NW, N or NE aspect. Approximately 40% of the catchment present SA values lower than 0.6 (Figure 3c). The elevation band with the largest values of SWEmax is that between 4500 and 5000 m a.s.l. with a S-SE aspect (Figure 3i), which correspond to the areas close to Tapado Glacier (around 10% of the catchment area, Figure 3g-i). These areas also have a very consistent interannual behavior (Figure S5). During the melt period, the catchment was mostly snow-free (Figure 3d), except for sites mostly above 5000 m a.s.l. with a SW-W aspect (Figure 3e), and only about 10% of the catchment area present values above 0.2 (Figure 3f).

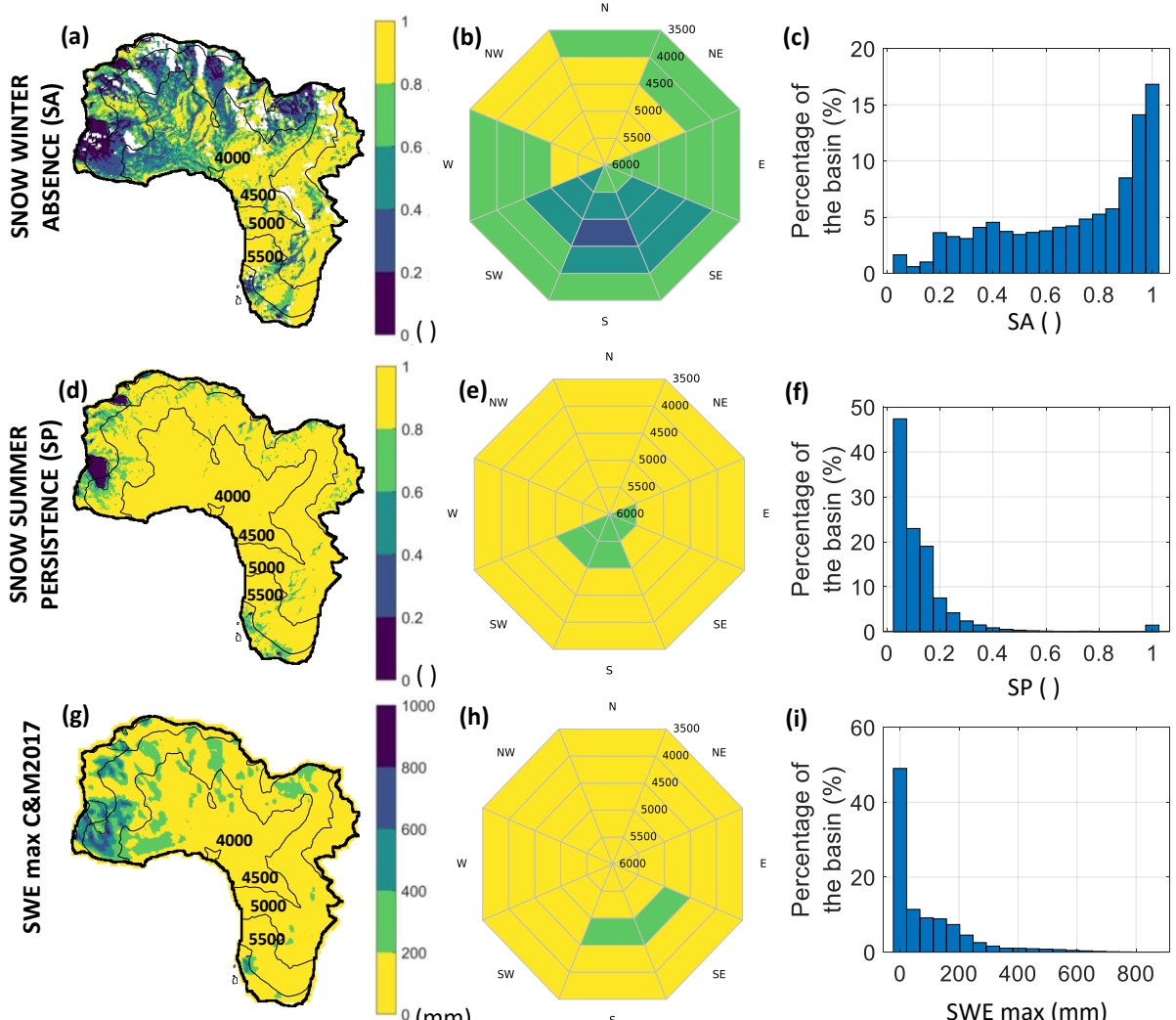

Figure 3: Snow winter absence (SA), snow summer persistence (SP) and the interannual median of peak annual SWE (SWEmax) in the Corrales basin. While SA and SP are derived following Wayand et al. (2018), SWEmax is calculated from the Cortés and Margulis (2017) dataset. (a) Map of SA, (b) Polar plot of SA, (c) Histogram of SA, (d) Map of SP, (e) Polar plot of SP, (f) Histogram of SP, (g) Map of SWEmax, (h) Polar plot of SWEmax, (i) Histogram of SWEmax. In the polar plots (b-e-h) the angles and the inverse radial distance represent the aspect and the elevation, respectively. While SA and SP refer to the percentage of time when snow is absence or present (a-b and d-e) in the accumulation (April-September) and melt (October-March) periods, respectively, the histogram represent the spatial distribution of the variables across the catchment. Glaciers outlines and contours are shown in (a), (d) and (g). Blank areas in (a) represent sites where the SA index cannot be calculated due to an insufficient number of cloud-free images in April-September.




## 5.2 Verification of SnowModel results

We verify SnowModel results by comparing them against the available field data and the snow products analyzed in Section 5.1. At the point scale, snow depth and SWE records at TAP compare well against the ensemble runs for the corresponding grid cell (Figure 4). The simulated changes of snow depth following precipitation events are similar to observations, particularly during periods of rapid depletion following each precipitation event. Rapid depletion periods are likely due to compaction, as well as mechanical removal and sublimation caused by frequent strong wind gusts and rapid drops in humidity occurring post precipitation. We discarded the possibility that these drops are caused by melt because air temperature was mostly below zero following events. We observe differences in the modeled and observed peak snow depth and SWE that are likely caused by discrepancies between the precipitation amounts registered by the precipitation sensor and the snow records. The snow disappearance dates are well approximated in 2019 (October) and 2020 (September and October) in the snow depth record (Figure 4a), but the SWE observations show a slower disappearance rate (Figure 4b).

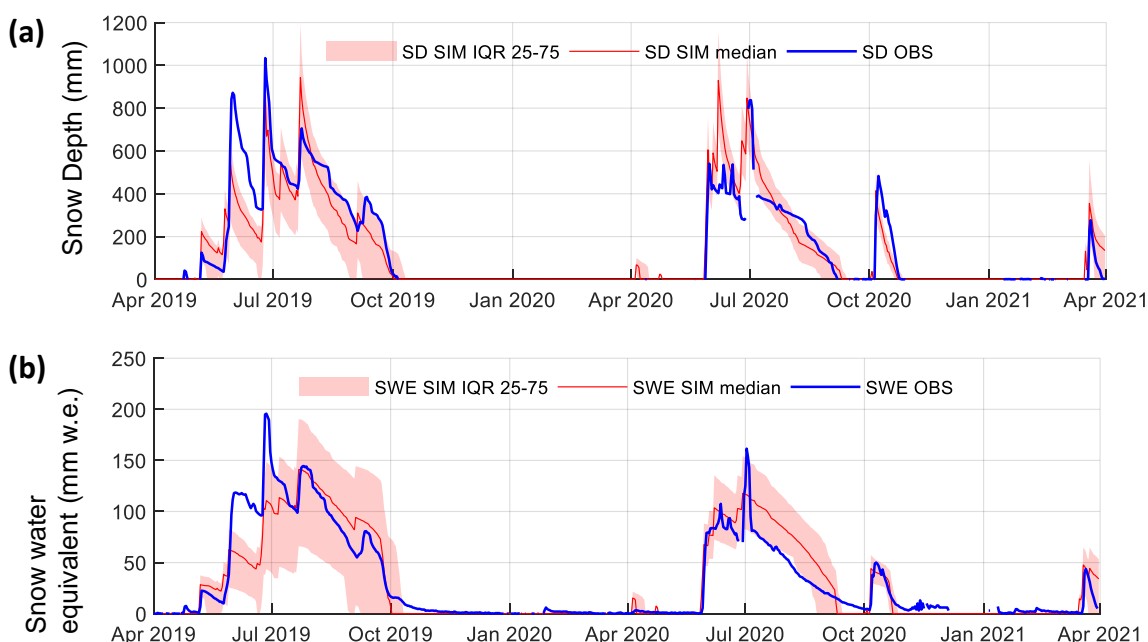

**Figure 4: Time series of the observed snow variables at TAP and the ensemble runs for the corresponding grid cell. (a) Snow depth, (b) SWE. The red lines and areas represent the median values and interquartile ranges of the SnowModel ensemble runs, respectively. The blue lines correspond to the observations.**

When we compare the SnowModel ensemble runs against the selected snow products (Sections 3.2 and 5.1) we find that our simulations correctly reproduce the main spatial patterns of the snow cover throughout the year, but tend to overestimate the winter snow cover area and smooth the spatial variability of snow during summer. The SCAs from the ensemble runs (red areas and line) coincide well with the Sentinel-2A SCA values (blue points) at several times throughout the study period (obtaining scores of $R^2=0.66$ and RMSE=14.8%), but the snow cover area is sometimes overestimated in winter and tends to

disappear too early in the simulations (Figure 5a). Figure 5b-c show the comparison between SA and SP derived from satellite images and from SnowModel. The SnowModel values of SA and SP follow the same definitions of the satellite-derived indices but using the results of the simulations. With respect, to the polar plots presented in Figure 3b-e-h, each point in Figure 5b-c-d represents the SA, SP and SWEmax average value of all the grid cells contained in a 500-m elevation band (from 3500 to 6000 m, 5 values in total) and 45° aspect (from N to NW, 8 values in total), whereas the error bars represent the standard deviation over the same grid cells. There is a relative good correspondence between observed and simulated values ($R^2$=0.47 for SA and $R^2$=0.67 for SP), but we observe a bias in SA (BIAS = −0.26) and an offset of approximately 0.5 in SP. Both effects could be explained by snow-free sites across the catchment that are not well reproduced by the model. An extreme case of this situation are the two data points that do not align to the general linear relationship of SA (lower right corner in Figure 5b), which correspond to sites located above 5500 m a.s.l. with an E and SE aspect. Figure 5d shows the same plots but comparing SWEmax (Cortés and Margulis, 2017). While the reference SWEmax is calculated from the 1985-2015 period, simulated SWEmax is calculated using the two hydrological years of the modelling period (2019-2020 and 2020-2021). As the reference and simulated datasets correspond to different study periods, this comparison is evaluated mostly in relative terms. The $R^2$ obtained by comparing both records is lower than that obtained for the snow indices ($R^2$=0.23), likely due to difficulties of SnowModel to reproduce the shallow and deep snowpacks produced by the reference dataset.

Finally, we also verifiy that the results of SnowModel are in good agreement with the set of ablation stakes summer readings collected on Tapado Glacier (Figure S6). Although there are differences between observed and simulated values, the main trends throughout the season are similar, with a higher ablation rate between November and February and a reduced rate after March.

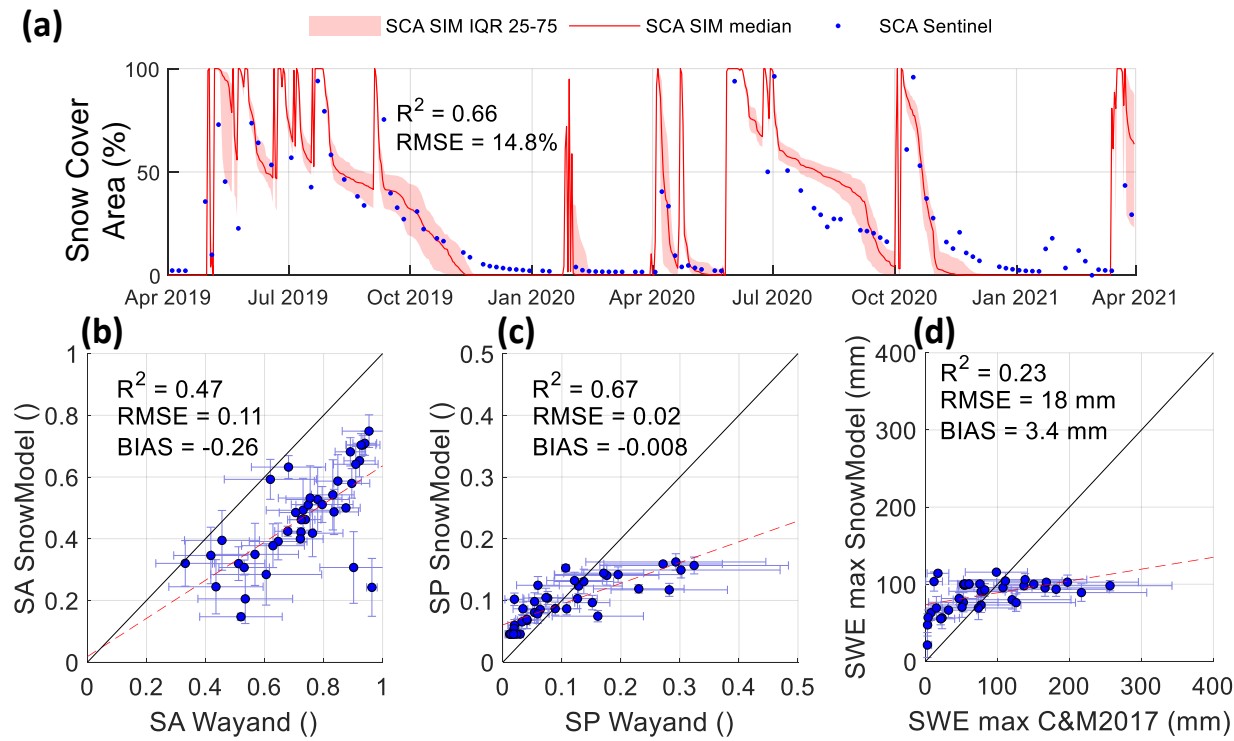

**Figure 5: Comparison of reference (Wayand and C&M2017) and simulated (SnowModel) snow variables at the scale of the Corrales catchment. (a) Snow cover area (SCA), (b) snow winter absence (SA), (c) snow summer persistence (SP), and (d) interannual median of peak annual SWE (SWEmax). In (a) the red lines and areas represent the median values and interquartile ranges of the SnowModel ensemble runs, respectively. The blue dots correspond to the observations. In b-d the blue points and error bars represent the mean and standard deviations values, respectively, within a 500-m elevation band (from 3500 to 6000 m a.s.l.) and a 45° aspect range (from 22.5° to 337.5°) as the polar plots in Figure 3b-e-h. As evaluation metrics we use the coefficient of determination ($R^2$), the root mean square error (RMSE) and the mean bias (BIAS).**

### 5.3 Snow mass balance and runoff generation

In this section we address the main objectives of the study by using the SnowModel simulations to quantify the snow mass balance and to describe the spatial distribution of snowmelt and sublimation fluxes. The catchment-average snow mass balance derived from the SnowModel ensemble runs is shown in Figure 6. Precipitation occurred mostly in autumn and winter (April to June) with some extraneous events in September 2019, January 2020, October 2020 and March 2021. Estimated rainfall was much lower than snowfall and was mostly restricted to summer and autumn. Snow surface sublimation was the process that removed most of the snow mass and dominated ablation during winter and spring. Snowmelt runoff and ice melt played a secondary role at the annual level, but their relative importance increased in spring and summer, respectively. From November to March, ice melt was almost the only runoff contributor. Fluxes derived from the transport of snow (wind-transported snow and blowing snow sublimation) were smaller than surface sublimation and took place only during winter or other periods with

fresh snow on the surface. While blowing snow sublimation is always a mass loss (negative values), wind-transported snow can add or remove mass to the catchment (positive or negative values), but in the study period it was mostly a mass loss. The inflow to La Laguna reservoir is shown in Figure 6 to understand the response of streamflow to snow and ice melt, but as the reservoir is located approximately 24 km downstream of the Corrales outlet it also includes the contribution from other sub-

catchments. We observe a certain correspondence between snow and ice melt from Corrales and La Laguna inflow in summer 2020, but this is not that evident in summer 2021. An interesting feature is that the dry winter of 2020 resulted in a notoriously lower streamflow in the next spring and summer.

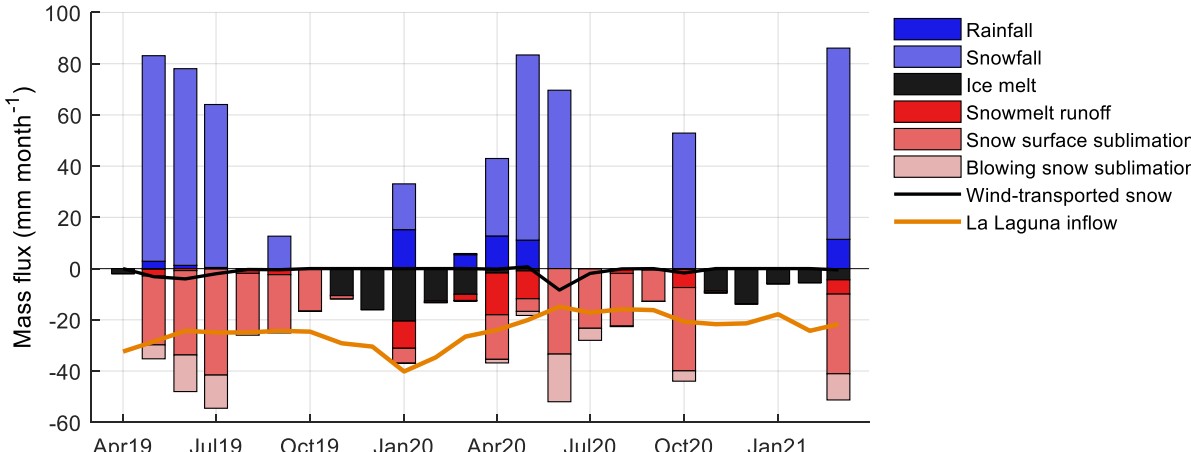

**Figure 6: Results from SnowModel showing monthly mass fluxes in the Corrales catchment as stacked bars. The bars**
**correspond to the median value of the ensemble runs. Liquid and solid precipitation are considered as inputs to the catchment (plotted as positive bars), whereas ice melt, snowmelt runoff, surface sublimation and blowing snow sublimation are considered as losses from the catchment (plotted as negative bars). Wind-transported snow is the sum of suspension and saltation and can be both an input and an output (plotted as a black line). The inflow to La Laguna reservoir (normalized by the Corrales catchment area) is plotted as a reference for streamflow variations in**
**the study region, but it includes the contribution from other sub-catchments downstream of the Corrales outlet.**

Next, we analyze the spatial distribution of snowmelt runoff and other fluxes that remove mass from the snowpack. During winter, snowmelt is minimum, and the snowpack loses mass mostly through sublimation wind transport. We find that the spatial heterogeneity of wind-related snow fluxes is large, with low values of wind transport and blowing snow sublimation on the north-west section of the catchment and larger values to the east. This is explained by the contrasting wind exposure of

western and eastern areas (see wind speed records in Figure 2). On the east of the catchment, snow erosion dominates over snow deposition (positive values), ranging from negative to positive values, with a large small-scale variability (Figure 7a). On average, snow erosion is largest at sites between 4500 and 5500 with a W-NW-N aspect (Figure 7c). Blowing snow sublimation, on the other hand, varies from low to large values from west to east, reaching up to 100 mm a$^{-1}$ above 4500 m a.s.l. at the border with Argentina (Figure 7b). The elevation bands that present the largest values of blowing snow sublimation

are mostly located above 4500 m a.s.l. and have a N-NE aspect (Figure 7d).

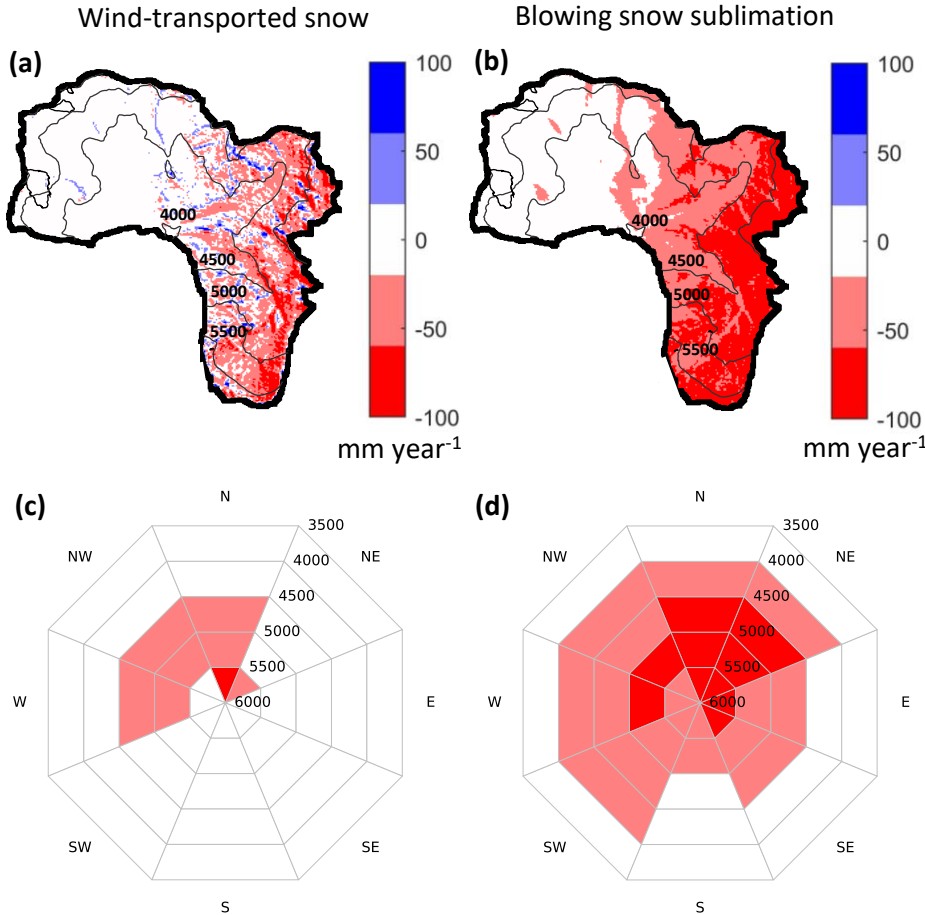

**Figure 7: Annual snow mass fluxes associated with wind transport calculated as the medians of the SnowModel ensemble runs. (a) Map of wind transport, (b) Blowing snow sublimation, (c) Polar plot of wind transport and (d) Polar plot of blowing snow sublimation.**


Once the snow metamorphism starts in response to internal exchanges of energy and vapor, wind transport is reduced, and the snowpack is more favorable to surface sublimation and snowmelt runoff. We find that surface sublimation was the biggest loss of snow mass, showing large values at high-elevation sites on the western areas of the catchment (Figure 8b). Snowmelt runoff shows a heterogeneous distribution with large values at wind-protected valleys in the north-west section of the catchment and

very low values to the east (Figure 8a). On average, the elevation band with the largest values of snowmelt runoff was that located between 4000 and 4500 m a.s.l. with a SE aspect (Figure 8d). The sublimation fraction is above 60% across the entire domain, and above 80% at high-elevation sites on the west, where surface sublimation is very large, and to the eastern areas of the catchment, where snowmelt runoff is almost zero (Figure 8c). Glaciers appear as sites dominated by snow surface sublimation with a sublimation fraction larger than 80% (Figure 8c). However, in terms of runoff volume, we find that ice melt

corresponds to 60% of the runoff contribution from the cryosphere (snowmelt and ice melt), which is equivalent to 55 mm a$^{-1}$ (or 4.3 Mm$^3$ a$^{-1}$, about 10% of the maximum capacity of La Laguna reservoir).

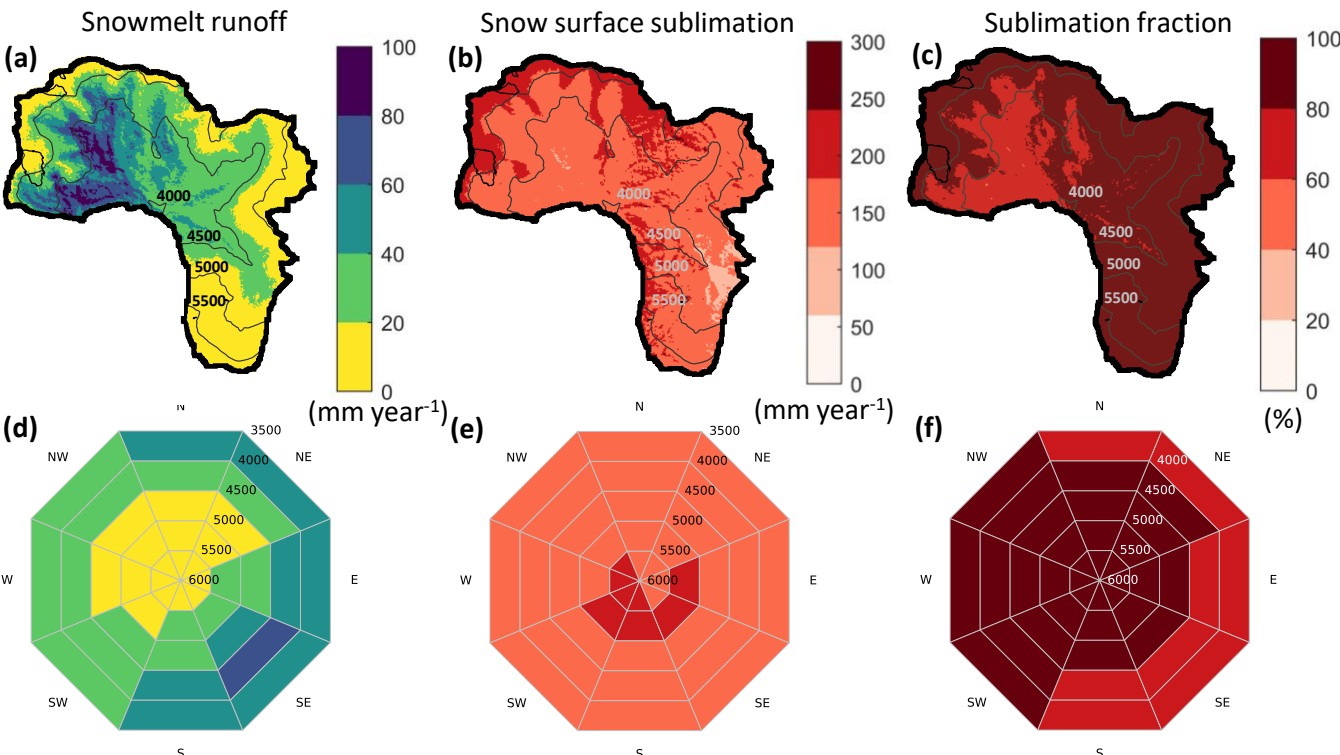

**Figure 8: Maps and polar plots of snow mass fluxes calculated as the medians of the SnowModel ensemble runs for 2019-2021. (a) and (d) Snowmelt runoff, (b) and (e) Snow surface sublimation, (c) and (f) Sublimation ratio.**


The contrast between runoff generation from (debris-free) glaciers and the rest of the catchment is further analyzed in Figure 9. In this figure, we compare snowmelt runoff and snow surface sublimation over the catchment and its glaciers using boxplots that explicitly consider the uncertainty of the simulations (SnowModel ensemble runs). Snow surface sublimation is the largest mass loss from the catchment reaching values between 100 - 250 mm a$^{-1}$ (Figure 9a) and represents between 71 - 90% of total

ablation (Figure 9b). Over glaciers, surface sublimation is larger than over the rest of the catchment, but snowmelt runoff is lower (Figure 9a), resulting in a very large sublimation fraction (Figure 9b). We find that snowmelt runoff at the catchment level and ice melt have similar mean values (between 34 and 55 mm a$^{-1}$), but the uncertainties derived from our ensemble runs are much larger for snowmelt runoff than for ice melt (Figure 9a).

The relationship between snowmelt runoff and snow surface sublimation to the selected inputs in Table 3 is assessed using an

R-squared analysis of the ensemble runs (Table 4). Ensemble variability in snowmelt runoff is mostly explained by $z_0$ ($R^2$=0.72) and less so by precipitation ($R^2$=0.11). Snow surface sublimation ensemble variability presents an almost opposite behavior being mostly explained by precipitation ($R^2$=0.65) and less so by $z_0$ ($R^2$=0.29). Interestingly, we find that snowmelt runoff is better explained by $z_0$ than snow surface sublimation is. In fact, as snow surface sublimation is the largest flux in the

snow mass balance of the catchment, this variable is mostly explained by total precipitation. Sublimation fraction is largely
dependent on $z_0$ ($R^2$=0.85), which controls the magnitude of turbulent latent heat fluxes.

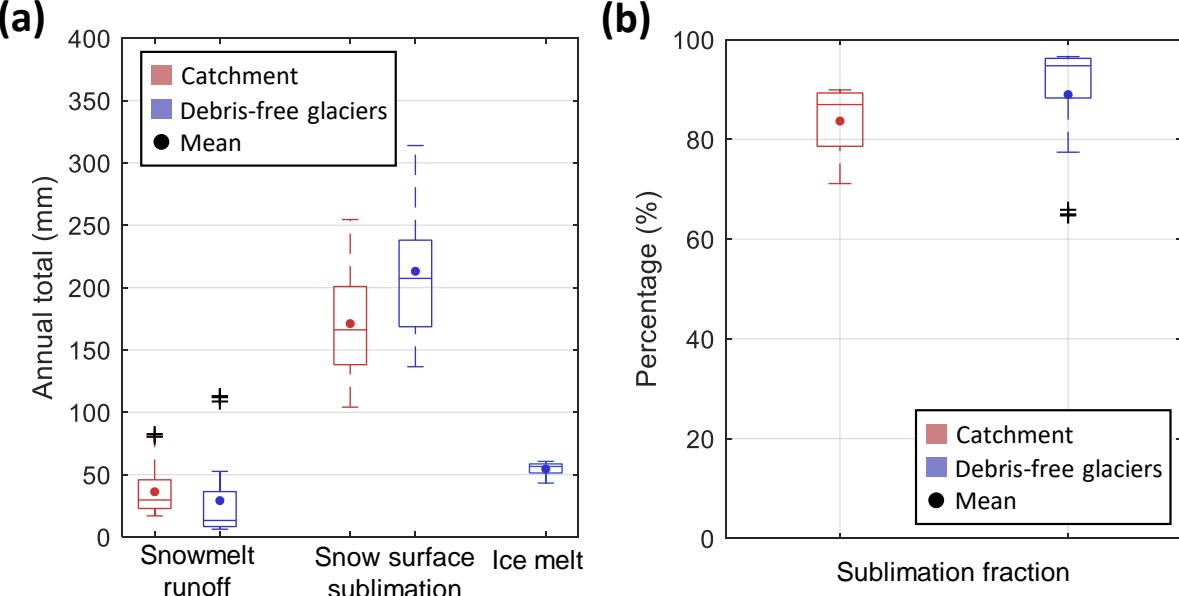

**Figure 9: (a-b) Boxplots of snowmelt runoff, snow surface sublimation, ice melt and sublimation fraction in the Corrales catchment (red boxplots) and debris-free glaciers (blue boxplots). In (a) ice melt values are normalized by**
**the catchment area. The edges of the boxes represent the 25th and 75th percentiles, the horizontal lines represent the median and the dots corresponds to the mean. Outliers are shown with a black cross.**

**Table 4: R-squared values of linear regressions between SnowModel parameters and outputs**

| | | Model inputs | | |
| --- | --- | --- | --- | --- |
| | | Precipitation factor | Aerodynamic surface roughness length for snow ($z_0$) | Wind factor |
| Model outputs | Snowmelt runoff | 0.11 | 0.72 | 0.00069 |
| | Snow surface sublimation | 0.65 | 0.29 | 0.00019 |
| | Sublimation fraction | 0.03 | 0.85 | 0.00039 |

**5.4 Snowmelt hotspots**

We further assess the spatial distribution of runoff generation based on the existence of sites that generate most of the snowmelt runoff in the study catchment. To depict the large heterogeneity of snowmelt runoff we plot the cumulative percentage of snowmelt runoff as a function of the cumulative area (Figure 10a). We find that 50% of the snowmelt runoff is generated from

21-29% of the catchment area (Figure 10a), which we label as snowmelt hotspots (Figure 10b). For this, we first rank the grid cells based on their snowmelt runoff and then we define as snowmelt hotspots the first grid cells of the ranking that produce 50% of the total snowmelt runoff. We also find that 50% of the catchment produces ca. 80% of the total snowmelt runoff (Figure 10b). This heterogeneity is larger than that of the annual peak SWE (blue areas and line in Figure 10b) and is an indication that snow removal by wind transport and sublimation further increases the spatial variability of the resulting seasonal snowmelt runoff. In Figure 10b, we show a map of the snowmelt hotpots calculated as the minimum area that produces 50% of the snowmelt runoff in the catchment (calculated from the median of the SnowModel ensemble runs). Interestingly, despite the large runoff contribution of ice melt, none of the debris-free glaciers are located on areas labeled as snowmelt hotspots.

To determine which factors control the location of snowmelt hotspots, we compare the cumulative distribution functions of several descriptive variables of snowmelt hotspots and the rest of the catchment (Figure 11). We find that snowmelt hotspots are located below 5000 m a.s.l., (mostly between 4200 and 4800 m a.s.l., Figure 11a), have a E, NE and SE aspect (ca. 80% of the snowmelt hotspots have an aspect lower than 180°, Figure 11b) and lower slope angles than the rest of the catchment (Figure 11c). In terms of snow variables, snowmelt hotspots show lower values of SA (Figure 11d), but relatively similar values of SP (Figure 11e) in comparison with the rest of the catchment. For this analysis, simulated SA and SP were calculated using every timestep in the simulation period (in contrast to Figure 5 where we use only the timesteps with available satellite images). Snow depth values at the time of maximum accumulation was of at least 0.7 m (Figure 11f).

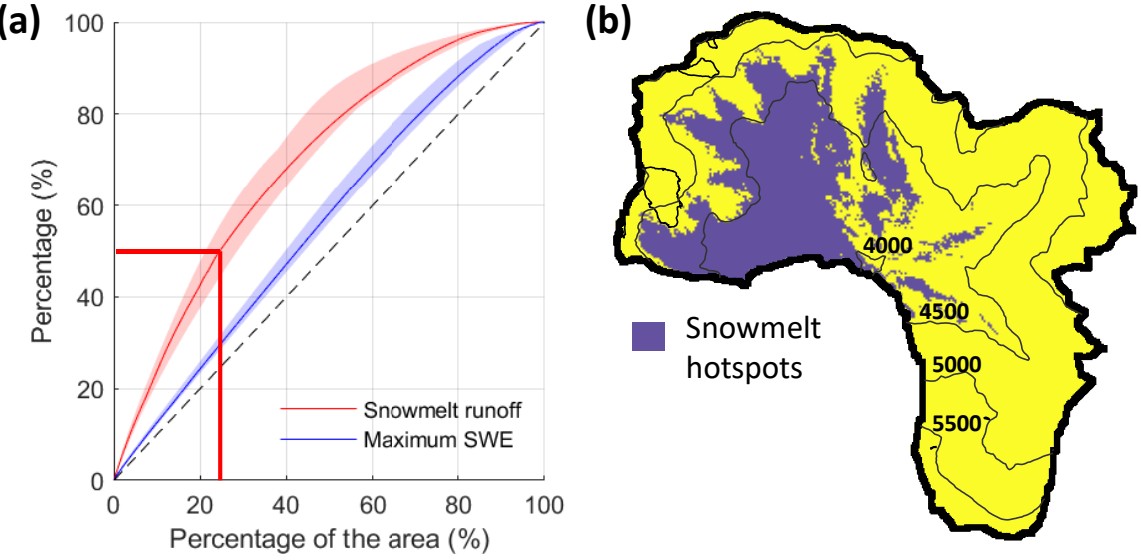

**Figure 10: (a) Cumulative simulated snowmelt runoff and maximum SWE against cumulative area, (b) Map of snowmelt hotpots. In (b) the red lines and areas represent the median values and interquartile ranges 25 and 75 of the SnowModel ensemble runs, respectively.**

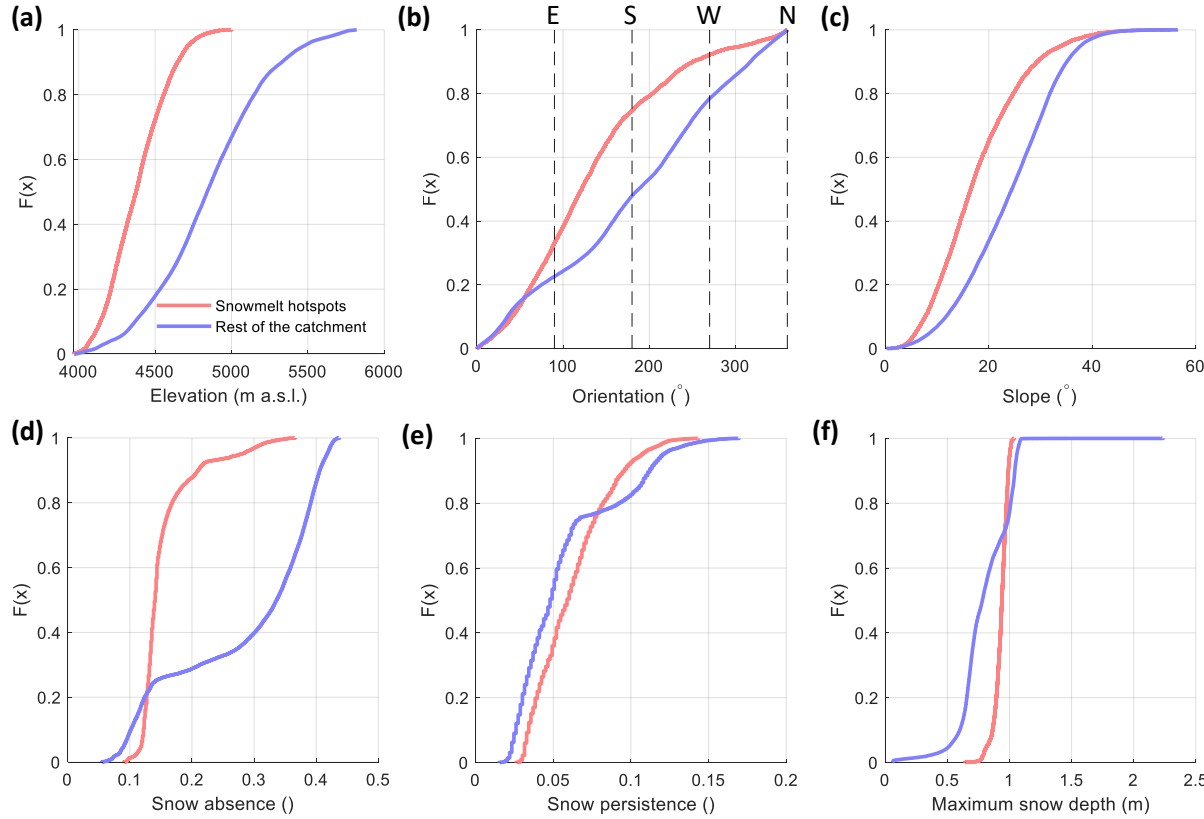


**Figure 11: Cumulative distribution functions (CDFs) of selected topographic and snow variables for snowmelt hotspots and the rest of the catchment (except hotspots). In upper part of (b) we add cardinal directions in letters for reference. (d) and (e) were made using the simulated SA and SP (f) was made using the ensemble median of the snow depth at the time of maximum accumulation.**


## 6 Discussion

### 6.1 Verification of model results

The comparison between simulated and reference datasets, both at the point and distributed scales suggest that our SnowModel ensemble runs correctly simulate the most important features of the snow cover in the Corrales catchment, but we identify a number of issues that we discuss in this section. At the point scale, the seasonal amounts of snow observed at TAP are well reproduced by SnowModel, but the simulations fail to match the magnitude of some of the accumulation peaks, particularly

in terms of snow depth during winter 2019 (Figure 5a). Additionally, most of the depletion curves are well captured by the model simulations, except for the depletion curve in the spring 2020, which does not follow the same rates shown by the snow

depth and SWE (Figure 5a-b). As suggested by Voordendag et al. (2021) when analyzing the TAP station site in winter 2017, the differences in precipitation peaks and depletions rates are likely given by the uncertainty in the precipitation forcing and the parametrization of fresh snow density and albedo. As Snowmodel does not consider the advection of sensible and latent

heat from snow-free areas, we suggest that this might be an additional factor that explain differences between the observed and simulated snow depletion rates (Mott et al., 2011). Interestingly, model results reproduce the main patterns of glacier ablation on the tongue of Tapado Glacier (Figure S5), although more detailed parameterizations are certainly needed to simulate the energy and mass balance on penitentes fields in their full complexity (Lhermitte et al., 2014; Sinclair and MacDonell, 2015; Nicholson et al., 2016).

Model results are also in good agreement with distributed datasets in terms of general spatial trends, but simulations show some differences with the snow products that are likely related to an insufficient small-scale spatial variability of simulated snow variables. There is a general underestimation of SA that can be explained by the simulated accumulation of snow at wind-exposed sites where snow is quickly removed (Figure 5b), and there is also a very large spatial variability of SP that is not reproduced by the model (Figure 5c). We find that the snow indices proposed by Wayand et al. (2018) are particularly

useful to evaluate our results as they provide additional information on the spatial distribution and duration of the snow, in contrast to analyses based solely on the percentage of SCA in the study basin. The $R^2$ obtained for the observed and simulated SA is lower than that obtained for SP ($R^2$=0.47 versus $R^2$=0.67, Figure 5b-c). This result is somehow expected because SA is controlled by accumulation processes that are difficult to reproduce in snow models, such as wind fields over complex terrain and preferential deposition (Mott et al., 2010; Freudiger et al., 2017; Hock et al., 2017). When comparing outputs from the

Canadian Hydrological Model (CHM) against satellite-derived SP, Vionnet et al. (2021) obtained evaluation metrics that are similar to those in this study (Pearson correlation coefficients between 0.69 and 0.75). The poorer performance of the model in the accumulation season can also be observed in the weak relationship ($R^2$=0.23) between the annual maximum accumulation from the model and the SWE reconstruction dataset that we use as reference (Cortés and Margulis, 2017). This weak relationship is in part explained by the large spread of simulated maximum SWE values when compared against low

values of the reference dataset (Figure 5d). This is likely caused by the simulation of low wind speeds and large snow accumulation at sites dominated, in reality, by snow erosion and high wind speeds. On the other hand, a correct representation of SP is more easily achieved because this index is more related to ablation processes that depend on variables that are easier to reproduce, such as solar radiation and air temperature gradients (Wayand et al., 2018). However, despite the high $R^2$ values obtained from the comparison of observed and simulated SP, the snow cover disappears earlier in the simulations than in the

satellite images (Figure 5a). This is likely caused by insufficient snow accumulation, or an accelerated ablation simulated by the model at high-elevation sites, whereas in reality snow can remain on the surface until the start of the summer (December). A key advantage of our study is the use of well-distributed AWS data to force the snow simulations, particularly the contrasting wind records of PAN and the rest of the AWSs, which are located at wind-exposed and wind-sheltered locations, respectively. These contrasting wind records allow the simulation of very different snow conditions at western and eastern areas of the

catchment. In fact, some differences between our simulations and those of Réveillet et al. (2020), who also used AWS forcing

in the same catchment, might be caused by the gaps in PAN records during their study period. For example, we obtain much larger sublimation ratios (~85% versus ~35%). Similarly, Gascoin et al. (2013) obtained a sublimation fraction of ca. 70% applying SnowModel to an instrumented site in Pascua-Lama (2600–5630 m a.s.l., 29°S) over the 2008 winter. Another good indication for our simulations are the results of point-scale studies on glaciers of the semiarid Andes, such as those from Ginot et al. (2006) and MacDonell et al. (2013), which obtained year-round sublimation ratios around 80% on the upper areas of Tapado and Guanaco (5324 m a.s.l., 29.34°S, 70.01°W) glaciers, respectively. On the other hand, Ayala et al. (2017b) obtained a sublimation fraction of 12% on the tongue of Tapado Glacier, but those results were obtained from a two-month summer period.

## 6.2 Runoff generation and snowmelt hotspots

In relation to the existence of snowmelt hotspots in the semiarid Andes, we have identified the areas that produce half of the snowmelt runoff in the Corrales catchment, and described their main topographic and meteorological characteristics. It is highly likely that our model simulations based on ensemble runs correctly identify the general location of the snowmelt hotspots, but as the model has difficulties to simulate the full small-scale variability of the snow cover (see Section 6.1), the exact location of the sites producing the largest snowmelt amounts is difficult to find. We estimate that these areas correspond to 21-29% of the catchment area. Although we use a threshold of 50% to define the snowmelt hotspots, alternative definitions might be applied in the future depending on the objectives of each study. These results are in line with other studies showing how the heterogeneity of snow processes affect runoff generation (DeBeer and Pomeroy, 2017). The hydrological relevance of snowmelt hotspots can also be connected with the results from Badger et al. (2021), who used idealized simulations to show that the heterogeneity of the snow cover delays snowmelt runoff and creates areas of snow persistence. We suggest that in other arid or semiarid catchments, snowmelt hotspots could be expected for leeward slopes with relatively low slope angles and moderate elevations. However, to precisely identify snowmelt hotspots and quantify their contribution in other regions, a modeling approach combined with satellite information should be applied such as the one presented in this work.

Snowmelt hotspots in the semiarid Andes are a clear illustration of the large spatial variability of physical processes (from accumulation to heat exchange) that ultimately control snowmelt runoff (Mott et al., 2018). First, the sheltering and exposure to strong winds during and after snowfall events determine large differences in snow accumulation, from areas of preferential deposition, such as glacier accumulation zones, to large areas of strong snow erosion by the wind and shallow snowpacks (Figure 3). When snow surface conditions are favorable to wind transport, snowdrift and blowing snow sublimation can mobilize and remove large amounts of snow mass from the catchment (Figures 6 and 7). These processes are not spatially uniform and can define areas of snow removal, such as the eastern section of Corrales. Second, the high surface sublimation rates (of up to 300 mm a$^{-1}$, Figure 8) remove snow and use large amounts of energy from the snowpack that would be otherwise available for melt. The combination of these two factors define an heterogeneous snow cover in which snow-free areas appear very early in the melting season (ca. 50% of the catchment area in August, Figure 6a), likely contributing to snowmelt through the advection of sensible and latent heat (Liston, 1995; Mott et al., 2020; van der Valk et al., 2022). A logical step towards

improving our estimates of snowmelt runoff in this type of environment would be the calculation of the contribution of local

heat advection from snow-free areas to snowmelt rates, but with a focus on turbulent latent heat fluxes. Other studies have estimated a contribution from heat advection to melt of up to 40% in alpine, prairie and Arctic environments (Mott et al., 2018). The removal of snow by sublimation processes is thus a key feature of dry mountain environments as it adds heterogeneity to the resulting spatial distribution of the seasonal snowmelt runoff. If sublimation rates were lower, the spatial distribution of the seasonal snowmelt runoff are expected to be very similar to that of the end-of-winter SWE, although the

snowmelt timing can be very variable from site to site due to differences in melt rates. According to our simulations, snowmelt hotspots are located at sites dominated by preferential deposition, reduced snow transport and low sublimation rates in relation with other wind-exposed areas of the catchment.

The existence of snowmelt hotspots is relevant for hydrological modelling and water management, since not all sub catchments with snow cover during winter contribute the same way to available water downstream during e.g. the irrigation season. Future

snow monitoring driven by the development of seasonal streamflow forecasts in the semiarid Andes could benefit from considering what are sites with the largest snowmelt contribution, because the relationships between streamflow and snow accumulation will be more sensitive at those sites. A better understanding of the factors driving differences between snowmelt hotspots and other areas could be achieved by installing meteorological stations that allow for the calculation of all the energy balance components.

Snowmelt hotspots might play a key hydrological role in connection with other components of this landscape, such as peatlands, groundwater recharge and rock glaciers. Valois et al. (2020) investigated the hydrological dynamics of a mountain peatland in a catchment close to Corrales and found that these features connect snowmelt at high-elevation areas with downstream agriculture and human needs, but there are no estimates of the runoff recharge from the melting of the cryosphere. In this direction, we recommend the assessment of the spatial connectivity between snowmelt hotspots and peatlands. The

infiltration of meltwater into rock glaciers is another possible link between snowmelt hotspots and the hydrology of the catchments (Schaffer et al., 2019; Pourrier et al., 2014). Recently, Navarro et al. (2023b) presented a set of geophysical measurements of the internal structure of Tapado debris-cover and rock glacier that reveal an intricate hydrological network that connects surface meltwater with massive ice lenses and ice-rich permafrost, but the dynamics of this connection is yet to be understood in detail. For example, it is not clear how the regulating role of rock glaciers changes in the presence of snow

rich or poor years. At a larger scale, Álvarez-Garretón et al. (2021) showed that the hydrological memory of snow-dominated catchments between 30 and 35°S are strongly influenced by the infiltration of snowmelt at high-elevation. Exploratory drillings have shown the presence of groundwater near the outlet of the Corrales catchment, which might be indicative of a recharge area. Finally, a brief analysis of satellite images acquired after the disappearance of the seasonal snow cover (Figure S7) suggests that snowmelt hotspots areas tend to have higher values of soil moisture and might contribute to the development of

riparian vegetation. In any case, this connection is not evident and needs to be verified and extended by field measurements and more in-depth analyses of remote sensing products.

Although glacier runoff is an important component in the water balance of the catchment and the semiarid Andes (Gascoin et al., 2011; Rodriguez et al., 2016), we found that the main glacier in our study catchment, Tapado Glacier, is not located in an area defined as snowmelt hotspot in this study. However, in line with the results of Gascoin et al. (2013), the snow products analyzed in this study and the results of the SnowModel simulations show that Tapado Glacier is located in an area of preferential deposition and low snow erosion rates (Figures 3 and 7). Furthermore, our results show that snow sublimation in glacierized areas is very high (Figure 9b), suggesting that sublimation and its associated turbulent latent heat fluxes significantly reduce summer ablation by reducing the energy available for melt. On the other hand, once the snow disappears, the ice surface is quickly melted (ice sublimation calculated by SnowModel was much lower than ice melt), creating large amounts of runoff that reach up to 60% of the runoff contributions in the catchment. These results confirm that glaciers are key contributors to runoff in this region, and the most important ones during dry periods. As the ice melt volume estimate of 4.3 $Mm^3$ $a^{-1}$ is relatively high in comparison with other studies (Robson et al. (2022) estimated that the average annual volume change of Tapado Glacier in the period 2012-2015 was of 0.6 $Mm^3$), we recommend the application of more detailed schemes to solve the energy and mass balance of glaciers in the Corrales catchment.

Although the advantageous location of the AWSs allowed us to capture the main patterns of the spatial variability of the meteorological forcing variables, the total number of stations is still insufficient to capture the full complexity of snow processes in this dry environment. Additionally, future modelling studies will benefit from the inclusion of heat advection from snow-free areas (van der Valk et al., 2022) and time-dependent parameterizations of $z_0$ (Réveillet et al., 2020; Voordendag et al., 2021). As the period analyzed in this study was very dry, the presence and role of snowmelt hotspots should be analyzed during wet years, where a more uniform distribution of snow accumulation could be expected. However, the hydrological processes that we analyzed in this study might become more frequent in the future, in line with the projections for this region in the context of climate change.

## 7 Conclusions

In this study we hypothesize and test the existence of areas providing most of the snowmelt runoff in a sublimation-dominated catchment in the semiarid Andes of Chile. For this we used a process-based snow model (SnowModel) to simulate the evolution of the seasonal snow cover over a two-year period in a 79 $km^2$ catchment (Corrales catchment) located in the upper areas of the semiarid Andes of Chile. Our simulations are analyzed using field data and a set of independent satellite-derived snow products that describe the spatial properties of the snow cover. Our main conclusions are as follows:

1.  Snow surface sublimation is the dominant snow ablation component across the study catchment, representing between 71 and 90% of total ablation from the surface (snowmelt plus snow surface sublimation). In winter and after snowfalls, blowing snow sublimation is also important and dominates over wind-exposed areas in the east section of the catchment.

2. We estimate that 50% of the snowmelt runoff is produced by 21-29% of the catchment area, which we define as "snowmelt hotspots". As in any mountain terrain, the large spatial heterogeneity of seasonal snowmelt runoff is a consequence of complex snow accumulation patterns, but in dry mountain environments, this heterogeneity is further enlarged by large sublimation rates, which reduce snow available for melt at wind-exposed sites.

3. Snowmelt hotspots in our study sites are mostly located at elevations between 4200 and 4800 m, have easterly aspects, low slope angles, and high snow accumulation.

As suggested by previous research in the semiarid Andes, glaciers of the study catchment play a key role during dry years, providing most of the runoff during the end of the summer. Snowmelt hotspots can be a useful concept to understand runoff generation in arid and semiarid mountain regions as it might help to identify the areas of hydrological connections between the components of the catchment, such as the snow cover, groundwater, peatlands and rock glaciers. Additionally, snowmelt hotspots are good candidates for the installation of hydro-meteorological stations, particularly in situations where only limited financial resources are available for the monitoring of large, remote regions. This study was carried out during a severe drought that affects the semiarid Andes since 2010. As current climate projections for this region suggest that droughts will be more frequent in the future, we expect that our results will be relevant in the context of climate change. In particular, they can be used to better understand the hydrological response of the semiarid Andes and to improve the planning and management of its water resources.

**Acknowledgments**

ÁA acknowledges ANID-Fondecyt postdoctoral project 3190732. We thank the project ANID-FONDEF IDeA I+D 2021 ID21I10129, ANID-CENTROS REGIONALES R20F0008 and all the people involved in the data collection on Tapado Glacier, particularly Ignacio Diaz. CEAZAmet is thanked for support with meteorological data acquisition. Gonzalo Cortés is kindly acknowledged for providing the SWE reconstruction dataset (Cortés and Margulis, 2017).

**Data availability**

Tapado and Paso Agua Negra meteorological raw data are freely available at www.ceazamet.cl. The dataset and scripts used in this study are available at 10.5281/zenodo.8029996.

**Author contributions**

ÁA designed the study with the collaboration of SM. ÁA led the collection at the Tapado Glacier field site. ÁA performed the SnowModel simulations and main analysis with the collaboration of SS. ÁA interpreted results and wrote the manuscript with the help of SM and SS.

## Competing interests

The authors declare that they have no conflict of interest.

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
