# Peer review of "Spatial distribution and controls of snowmelt runoff in a sublimationdominated environment in the semiarid Andes of Chile"

_Hydrology and Earth System Sciences, 2023_

## Author Comment (AC1)

**Reviewer 1**

**Review of: "Spatial distribution and controls of snowmelt runoff in a sublimation-dominated environment in the semiarid Andes of Chile" by Álvaro Ayala, Simone Schauwecker and Shelley MacDonell.**

This paper presents an interesting case study of a catchment in the Andes, which is a snowmelt-dependent region in which sublimation plays a significant role on the snow cover and water balance. The paper builds on previous studies focussing on modelling performance and underlying snow processes. The authors perform an elaborate analysis on the hydrological importance of the processes occurring in the Corrales catchment, Chile. In general, this is a well-written manuscript. However, parts of the manuscript require some additional attention, so that the overall quality of the manuscript improves. As such, I advise the paper to be revised before publication. Below I have stated more general and specific comments, which I hope the authors consider to be constructive.

We thank the reviewer for their positive comments and thoughtful suggestions. We have carefully revised the manuscript accordingly. Please see below our specific responses.

General:

Results:

The results contain a lot of information and figures, all of which are important. However, it is sometimes hard to make the connection to the other results for me as reader. Each figure is treated separately, and not always clearly connected to previous results. To illustrate, almost each paragraph starts with "In figure x, we compare…" or "Figure x shows …".

I would advise the authors to focus on the point you are trying to make and try to make in-text connections between the separate figures based on the general story. This results in a storyline in which the figures are a helpful tool instead of treating the results as a point-by-point discussion of the figures. Another option would be to merge the results in the discussion, however that is also not sufficiently done currently.

We appreciate your comment as it has allowed us to enhance and better organize our article. In the revised version, we have modified the text to highlight our main storyline, which is the connection between high snow sublimation rates and the spatial distribution of snowmelt in dry mountain environments. The spatial variability of total snowmelt runoff in mountain terrain is large due to the complex patterns of snow accumulation and snowmelt. While snow accumulation is controlled mostly by preferential deposition, wind redistribution and gravitational transport (Freudiger et al., 2017; Mott and Lehning, 2010), during the melt season the interplay between the surface energy balance components can create large spatial differences in snowmelt rates (DeBeer and Pomeroy, 2017; Pohl et al., 2006). In our article, we argue that the resulting spatial variability of total snowmelt runoff in the semiarid Andes is further enlarged by unevenly distributed large sublimation rates that greatly reduce the snow mass available for melt and define

relatively small areas that concentrate most of the snowmelt runoff. This response is also in line with the response to the second reviewer.

Figures:

All the figures used in the manuscript are important, and significantly contribute to the manuscript. However, multiple figures are rather unconventional. For example, some figures miss an x-axis and/or y-axis label or contain a strange diagonal line through the colorbar. Also, it seems that part of the figures consist of multiple loose figures, which are not all aligned. I encourage the authors to re-do part of their figures, so that these look more professional. (See the specific comments for examples).

Thanks for your detailed suggestions. We have followed them, and we have also restructured some of the figures to improve the communication of the article's main points. The main changes are:
- Figures 3 and 4a-b-c have been merged to simplify their message.
- Figures 4d-e-f and 7 have been moved to the Supplement as we decided that, although they present valuable information, they partially interrupt the main storyline.
- We have addressed all the specific comments regarding the figures. Please see our detailed responses.

Data and code:

I am happy to see that the data used in this manuscript can be found online. However, I highly encourage the authors to also publish their code used for the data analysis. This would make the research more align with the FAIR principles and also accessible for interested readers.

We are in the process of organizing and commenting on our main codes for interested readers. We will upload these codes as well as the main SnowModel outputs when uploading the revised version of the article.

Specific comments:

L71-74: The definition of snowmelt hotspots is not completely clear for the reader, especially when reading the paper for the first time. The second sentence could also refer to the areas where snow surface sublimation dominates over snowmelt.

In the revised version we are now more explicit and we have changed "these sites" to "the areas producing most of the snowmelt runoff".

L101 – 116 and Figure 1: Is discharge data also available? It seems like that based on Reveillet et al., (2020) (L92-94). This would be beneficial for the understanding of the reader, especially when discussing the hydrological importance. (Also later on in combination with Fig. 8)

The reference to Réveillet et al. (2020) in lines 92-94 is used only to back up the sublimation estimates (50-80% of annual snowfall). We have changed the wording to make this clear. In the revised Figure 6 (old Figure 8) we have included hydrological data corresponding to the inflow to

La Laguna reservoir (see below), which is located some kilometers downstream of Corrales outlet point.

[Figure]

L128: Could you briefly elaborate on what this simple method entails? In general, I agree that this method is in the Supplementary materials.
We have included the following sentence in the revised version: "The method is based on the identification of positive changes in the daily precipitation cumulative record that lead to increases in the 5-day moving average of the same series."

L224-225: Is there a specific reason why you do not consider rain-on-snow events in the snowmelt runoff variable? Previous studies have shown the significant effect rain-on-snow events can have on runoff. Based on Figure 8, I see that rain especially takes place in summer and autumn, during which temperatures are around 0 oC and snowfall also takes place, which could result in ideal conditions for rain-on-snow events to generate relatively high runoff, partly from the snowpack.
Thanks for this suggestion. We have decided to include rain-on-snow events in the definition of snowmelt runoff. In the revised version, snowmelt runoff consists of all runoff from the base of the snowpack, i.e. runoff originated from snowmelt and runoff originated during rain-on-snow events. We verified that after the inclusion of rain-on-snow events the ensemble median of snowmelt runoff increased by 4 mm a$^{-1}$ (from 69 mm a$^{-1}$ to 73 mm a$^{-1}$). More details have been added to Section 5.3. Some other numbers have changed in the manuscript (e.g. sublimation ratio) without modifying the main conclusions.

Tables 3 & 4: In these tables the input parameters are presented for the simulations. But it is unclear for me if this results in two "types" of simulations. Do you perform one base simulation (Table 3) and the ensemble runs (Table 4). Or do you vary the parameters in Table 4 as input in the simulations (Table 3)? In the former case I don't understand where you use this "base" simulation. If the latter, couldn't these tables be combined?
We thank the reviewer for noticing this. There is no "reference" or "base" simulation in our study. Table 3 was originally meant for such a case, but this was not included in the article. Tables 3 and 4 appear combined in the revised manuscript.

L245-246: Could you elaborate on the physical meaning of the slope and curvature wind distribution weights?

We have included this sentence in the text:

"The slope and curvature distribution weights increase wind speed in the presence of windward and convex slopes and decrease it in the case of leeward and convex ones (Liston et al., 1998)."

L261: Perhaps I misunderstood the definition of SP, but doesn't a value of 0.2 mean that there is only 20% of the time snow present during the ablation period? If that is the case 0.2 seems to me also mostly snow-free.

We agree with the reviewer that SP=0.2 is mostly snow-free. We reworded this sentence to: "During the melt period the catchment is mostly snow-free as revealed by the SP map (SP<0.2, Figure 3d).".

Figure 3: In the text, you refer to valley bottoms and ridges (L255), but it is hard to come to same conclusions based on your figures. Would it be an option to add isohypses to a and b? Additionally, I would advise to add labels to the colorbars, and add a y-axis label to the c and f figures.

We thank you for your suggestions. We have re-structured the figure and improved the visibility of the maps. Please note that we have merged Figure 3 with Figure 4a-b-c as these figures have similar patterns. Former Figure 4d-e-f- has been moved to the Supplement.

[Figure]

L269-270: I recommend to include the equation used to compute the coefficient of variation and explain how you compute these terms. This will leave no space for any uncertainties on how you computed these.
Thanks. We have included the equation (CV=standard deviation/mean) in the main document.

L285-294: The verification of the model simulations partly is performed based on a single observation site. The authors compare snow depth and SWE observed at Tapado with the modelled version of these variables representing the entire grid. Is there any evidence on how representative the measurements are for the entire catchment? How complex are the surroundings of that specific measurement site in relation to the entire catchment? Is the measurement site at a wind-exposed or wind-sheltered place?
We would like to note that TAP records are not compared against variables representing the entire grid but the variables representing the corresponding grid cell. We have modified the caption of the revised Figure 4 in case the text was not clear about this. In relation to the rest of the catchment, TAP is located in a wind-sheltered area where snow accumulates every winter. The

results of our paper about preferential snow accumulation on snowmelt hotspots also reinforce this idea.

L291: What do you mean with the Geonor sensor? I suspect that is the precipitation measurements based on table 1?
Yes, that is the precipitation sensor. We present it in Table 1, but it was not very noticeable in the first version. We have changed the reference in L291 from the "Geonor sensor" to "precipitation sensor".

L308-309: How do you compare the satellite-derived indices with the model-derived indices? Do you use the model values exactly at the moment of the satellite overpass? Or do you average the model values over a certain period?
Both the satellite-derived and the model-derived indices follow the same definition, i.e. original lines 184-185: "The snow absence (SA) index is defined as the fraction of time in which snow is absent during the accumulation period, whereas the snow persistence (SP) index is defined as the fraction of time in which snow is present during the melt period.". We followed the hydrological year to define the accumulation period as April-September and the melt period as October-March. While the satellite-derived indices are calculated using the times of image acquisition, the model-derived indices were originally calculated using every time-step in the corresponding periods. However, following the reviewers' suggestions, in the revised version we calculate the model-derived indices using only the image acquisition dates. This change has made observed and simulated values more similar in magnitude.

Figure 6: Are SA and SP Wayand the observations?
Yes, we interpret the satellite-derived indices as observations. We have added that to the caption.

Additionally, I would recommend to add a 1:1 line and the equation of the trendline, so it is clear that the absolute values do not match. Also out of curiosity, is there a reason why you do not force the fit through [0,0] (i.e. leave out the intercept). Theoretically, the simulations should be the same as the observations, so would justify removing the intercept.
Following the changes in the calculation of the model-derived SA and SP indices the absolute values match better than in the original version of the manuscript. We have added the 1:1 line. The intercept in the revised SA plot is almost negligible, but we prefer to keep the intercept as an indication of the offset in SP and SWEmax. We clearly acknowledge that in the revised manuscript. We have also included new metrics that help to better understand the relationship between the simulated and reference datasets: root mean square error (RMSE) and mean bias (BIAS). Please see the revised Figure 5 here.

[Figure]

L299-316: In this paragraph (also in the discussion), you refer multiple times to the R2 as correlation. Formally, R2 is the coefficient of determination and not the correlation. Yet, obviously, both are closely related. Additionally, the numbers in the text are not exactly the same as the numbers in the figures.

Thanks for noticing this. We have changed "correlation" to "coefficient of determination" to be precise. We have also double checked the numbers in the text.

Figure 7: What do the different markers mean? Am I correct to interpret these as different stakes?
Yes, each marker represents a different stake. We are now more explicit in the caption. Please note that this Figure has been moved to the Supplement.

Figure 9: I would advise to use the same colorscales for the maps and polar plots. Also, In the colorbars of the maps, some strange diagonal dashed line is present. Lastly, I suspect the caption is not complete.
Thanks for your suggestion. We have carefully revised the colormaps and made them consistent throughout the manuscript. We removed the diagonal bars and completed the caption.

Figure 10c: why is there a message in the figure? I agree that this is an important message, but this can also be inferred without the message (and is also stated in the text).
That is a result that we wanted to highlight, but we have removed it as requested by both reviewers.

Figure 12: it is hard to assess which areas are positive and which are negative, due to the chosen colorscales.

We have changed the color scales to facilitate the identification of negative and positive values (from red to blue with white for low absolute values).

Also, I suspect the caption is incomplete.

Thanks for noticing this. Actually, there is an error in the letters of each panel. Those letters referred to another figure. Please see the revised Figure 11.

[Figure]

L433-447: The authors start this paragraph by stating that the model results are in good agreement with the distributed datasets. I only partly agree with them. The R2 shows indeed relatively good scores, but this is not the case for the absolute values, which shows that the simulations underestimate the indices at least by a factor 2.

Since we have changed the calculations of the model-based snow indices, the reference and simulated absolute numbers agree much better than in the original version of the manuscript. We have included the RMSE and the mean bias as additional metrics to better describe the relationship between reference and simulated datasets.

I would recommend the authors to also mention the performance based on absolute values and put both these performances in perspective to previous studies. For example, is this known to be a common case with SnowModel?

To our knowledge, this is the first study that uses the indices proposed by Wayand et al. (2018) to evaluate outputs from SnowModel. However, the study by Vionnet et al. (2021) used the same indices to validate outputs from the Canadian Hydrological Model (CHM). Vionnet et al. (2021)

found that Pearson correlation coefficients of simulated snow depth and SP vary between 0.69 and 0.75, equivalent to $R^2$ values between 0.48 and 0.56, which is similar to the ones found in our study for observed and simulated SP (revised figure 5, see previous answer). In the case of SnowModel, Réveillet et al., (2020) and Voordendag et al. (2021) have analyzed the snow cover area in the same region. This variable (SCA) has been both under and overestimated and this was mostly attributed to the uncertainty in input data (Réveillet et al. 2020; Voordendag et al., 2021).

And is there an explanation for these mismatches in absolute values?

The mismatch in absolute values was corrected after the change in the calculation of the model-based snow indices.

L473-L485: This would be a nice place to discuss the dominant processes that you found in the Corrales catchment and what could be the cause of the snowmelt hotspots. However, you do not go into depth, and only briefly touch upon "the large spatial variability of the physical processes that control snowmelt runoff". I encourage you to elaborate more on what you found, which could serve as an overview of your findings merged into one story. Discussing this, would allow you to also compare your results with other regions in the world, especially where sublimation also plays a significant role.

In the revised version, we have extended this discussion to address the cause of snowmelt hotspots. We argue that the typically large spatial variability of snow accumulation and snowmelt rates in mountain terrain is further enlarged in dry environments by large sublimation rates that are unevenly distributed. These large sublimation rates almost completely remove snow cover from wind-exposed sites leaving very little snow available for melt. This discussion relates with the distribution of turbulent heat fluxes which has been addressed in other study areas, but with a more prominent focus on sensible heat fluxes than on latent heat fluxes. We have included and extended these points in the revised version.

L424-459: I miss a discussion on how well SnowModel generally performs based on the previous studies and how this could relate to your results. For example, could it be the case that SnowModel often overestimates snowmelt in specific parts of a catchment? A discussion on this would clarify whether you actually found snowmelt hotspots or are looking at the modelling uncertainty.

Thanks for this suggestion. We have included a more critical discussion on snowmelt hotspots and model uncertainty. In general, the literature available for this region has suggested that the uncertainty of input data has the largest impact on snow simulations over model selection and most of the model parameters (Gascoin et al., 2013; Réveillet et al., 2020; Voordendag et al., 2021). In our article we attempted to address this uncertainty by creating an ensemble of model runs based on three of the most uncertain parameters (precipitation, roughness length and wind factors). Based on these results, we can say that the heterogeneity of snowmelt and the presence of snowmelt hotspots are not modified within our uncertainty ranges (see revised Figure 9a below). The revised Figure 9b repeats the plot "percentage of the variable" against "percentage of the area" for the map of maximum SWE, showing that this variable is more uniform that total snowmelt runoff. Moreover, despite the uncertain input data in this region, we can be sure that

sublimation rates are large and they consume a large fraction of the snow mass available for melt at most sites, except at those identified as hotspots.

[Figure]

L486-488: It is unclear what you mean here? What part of the results do you refer to?

We were referring to the fact that in dry mountain regions sublimation removes large fractions of the snow mass, which would be otherwise available for melt. In these lines we had hypothesized that in more humid environments all snow would eventually melt. In the revised version, we have improved the wording.

**References**

DeBeer, C. M. and Pomeroy, J. W.: Influence of snowpack and melt energy heterogeneity on snow cover depletion and snowmelt runoff simulation in a cold mountain environment, J. Hydrol., 553, 199–213, doi:10.1016/j.jhydrol.2017.07.051, 2017.

Freudiger, D., Kohn, I., Seibert, J., Stahl, K. and Weiler, M.: Snow redistribution for the hydrological modeling of alpine catchments, Wiley Interdiscip. Rev. Water, 4(October), e1232, doi:10.1002/wat2.1232, 2017.

Gascoin, S., Lhermitte, S., Kinnard, C., Bortels, K. and Liston, G. E.: Wind effects on snow cover in Pascua-Lama, Dry Andes of Chile, Adv. Water Resour., 55, 25–39, doi:10.1016/j.advwatres.2012.11.013, 2013.

Liston, G., Sturm, M. H., En, G., Lr Ston, E. and Sturm, M. H.: A snow-transport model for complex terrain, J. Glaciol., 44(148), 498–516, doi:https://doi.org/10.3198/1998JoG44-148-498-516, 1998.

Mott, R. and Lehning, M.: Meteorological Modeling of Very High-Resolution Wind Fields and Snow Deposition for Mountains, J. Hydrometeorol., 11(4), 934–949, doi:10.1175/2010JHM1216.1, 2010.

Pohl, S., Marsh, P. and Liston, G. E.: Spatial-temporal variability in turbulent fluxes during spring snowmelt, Arctic, Antarct. Alp. Res., 38(1), 136–146, doi:10.1657/1523-0430(2006)038[0136:SVITFD]2.0.CO;2, 2006.

Réveillet, M., MacDonell, S., Gascoin, S., Kinnard, C., Lhermitte, S. and Schaffer, N.: Impact of forcing on sublimation simulations for a high mountain catchment in the semiarid Andes, Cryosph., 14(1), 147–163, doi:10.5194/tc-14-147-2020, 2020.

Vionnet, V., Marsh, C. B., Menounos, B., Gascoin, S., Wayand, N. E., Shea, J., Mukherjee, K.

and Pomeroy, J. W.: Multi-scale snowdrift-permitting modelling of mountain snowpack, Cryosphere, 15(2), 743–769, doi:10.5194/tc-15-743-2021, 2021.

Voordendag, A., Réveillet, M., MacDonell, S. and Lhermitte, S.: Snow model comparison to simulate snow depth evolution and sublimation at point scale in the semi-arid Andes of Chile, Cryosph., 15(9), 4241–4259, doi:10.5194/tc-15-4241-2021, 2021.

Wayand, N. E., Marsh, C. B., Shea, J. M. and Pomeroy, J. W.: Globally scalable alpine snow metrics, Remote Sens. Environ., 213(April), 61–72, doi:10.1016/j.rse.2018.05.012, 2018.

---

## Author Comment (AC2)

**Reviewer 2**

In the manuscript "Spatial distribution and controls of snowmelt runoff in a sublimation-dominated environment in the semiarid Andes of Chile", the amount of snowmelt and the snow sublimation are quantified for a catchment of 78 km2 using the SnowModel and two years of measured meteorological data. The paper aims to present the spatial distribution of snowmelt and snow sublimation processes and to do that, it defines so-called 'snowmelt hotspots' in the catchment. The study concludes that 50% of the snowmelt occurs in only around 20% of the catchment.

Overall, I found the study interesting, mostly well written and the topic well-suited for HESS. However, I have some doubts about the novelty of the study. It is not surprising that snowmelt is spatially heterogenous, and the paper also cites quite some studies that already looked at the contribution of sublimation exactly at this location. In the introduction it is written that these studies rather focus on models and uncertainty, rather than hydrological importance. However, in this study the 'hydrological importance' part is unfortunately not so clear either: only a short statement about the implication of snowmelt hotspots on recharge areas is given. I think the study would clearly benefit from describing more explicitly the added value of this study in the introduction, discussing in more depth the implications for snow science (in semi-arid areas) and explain more explicitly the hydrological importance of the findings.

We greatly thank the reviewer for their thoughtful comments. In the revised version, we have stated more clearly the novelty of this study based on the following points:
- The spatial variability of total snowmelt runoff in mountain terrain is large due to the complex patterns of snow accumulation and snowmelt. While snow accumulation is controlled mostly by preferential deposition, wind redistribution and gravitational transport (Freudiger et al., 2017; Mott et al., 2010), during the melt season the interplay between the surface energy balance components can create large differences in snowmelt rates across a certain domain (DeBeer and Pomeroy, 2017; Pohl et al., 2006).
- In the Introduction, we hypothesize that the resulting spatial variability of total snowmelt runoff in the semiarid Andes is further enlarged by unevenly distributed large sublimation rates that greatly reduce the snow mass available for melt and define relatively small areas that concentrate most of the snowmelt runoff.
- In the Discussion, we argue that this situation is distinctive of dry mountain ranges. In mountain ranges with low sublimation rates, the ablation of the end-of-winter snow cover is dominated by melt and consequently total seasonal snowmelt runoff is expected to have practically the same spatial distribution as snow accumulation, although the snowmelt timing can be very variable from site to site due to differences in the spatial and temporal distribution of melt rates.
- Additionally, we have added a more in-depth analysis of the implications of our findings for snow science and hydrology, such as the differences in the energy balance of snowmelt hotspots and the rest of the catchment; the role of wind in the exchange of turbulent fluxes that increase and decrease melt rates through sensible and latent heat fluxes, respectively; the difficulties to identify adequate snow monitoring sites; the possible connection between snowmelt hotspots and soil moisture and vegetation; and

the expected differences between the end-of-winter SWE and streamflow volumes at downstream locations.
- The hydrological significance of our findings has been further assessed based on two datasets: i) a comparison of snow and ice melt from the Corrales catchment against records of inflow to La Laguna reservoir (revised Figure 6), and ii) contrasting the location of snowmelt hotspots and soil moisture and vegetation in the catchment derived from satellite images (Figure S7). The first analysis shows the low snowmelt during the study period due to below-average precipitation and how most of the summer streamflow increase could be explained by ice melt. The second dataset suggests that the areas with the largest values of soil moisture and vegetation indices in the study period might be related with the location of snowmelt hotspots.

My other main concern is the presentation of the results and the figures. Sometimes units are not described, the same color bars for everything are confusing, text is added at strange places and captions are not always informative enough. The manuscript presents a lot of figures, which are sometimes only described in very few sentences in the results section and rather in a disconnected way. It would be helpful if the figures and the text together form a story and are answering a question or research gap that is presented in the introduction.
We thank the reviewer for the suggestions. In the revised version, we have corrected all the presentation problems identified by the reviewer. The disconnection between the text and figures was also raised by the other reviewer, and we have worked on the Results section to emphasize the general storyline. As a result, parts of the text have changed, and we have restructured some figures. Figures 4d-e-f and 7 have been moved to the Supplement as we decided that, although they present valuable information, they do not fit entirely into the main message.

Please see below for a point-by-point response including the new figures that will be presented.

Please find below more detailed comments:

L14: 'satellite-derived …….product' – suggest to leave out, because it comes also a few sentences later
We have removed the first use of "satellite-derived" to avoid repetition.

L17: 'absence and persistence' – maybe add the season?
We have reworded it to "snow winter absence and summer persistence". We have also added "winter" and "summer" to the labels in the revised Figure 3.

L19: here the characteristics of the snowmelt hotspots are shortly summarized. Maybe you could indicate which of these elements are likely also applicable to other regions/catchments.
We now describe snowmelt hotspots using characteristics that are more transferable to other regions: "Snowmelt hotspots are located at mid-to-lower elevations of the catchment on wind-sheltered, low-angle slopes".

The following sentence about "we suggest that snowmelt hotspots play a key hydrological role" is a too strong statement for the abstract as this is not shown in the current study. I suggest to reformulate.

We have reworded the last sentences of the abstract to "Our findings suggest that sublimation rates play an important role controlling the spatial variability of total snowmelt runoff in dry mountain areas. Snowmelt hotspots might be connected with other features of dry mountain regions, such as areas of groundwater recharge, rock glaciers and mountain peatlands, and recommend more detailed snow and hydrological monitoring of these sites, especially in the current and projected scenarios of scarce precipitation."

L41: shouldn't it be the other way around? i.e. decreases the energy and therefore lowers the temperature?

We have clarified changing these sentences to: "turbulent latent heat fluxes associated with the solid to vapor transition use energy from the snowpack, lowering its temperature and decreasing the energy available for melting".

L66: "From another perspective" – not clear what is meant here

We have changed "From another perspective" to "From a geostatistical perspective" and provide more a few more details about Mendoza's study: "From a geostatistical perspective, Mendoza et al. (2020) analyzed the spatial properties of a set of Lidar snow depth measurements across several catchments of central Chile and found a strong relation between snow depth and local topographic and land cover properties."

L116: here I wondered why the groundwater and hydrological data are not used in this study?

Unfortunately, the quality of the available hydrological data is still not that good to be included in the paper as some major gaps make the assessment and interpretation difficult. We have decided to change the last sentence to: Since 2009, the Corrales catchment has been instrumented with meteorological equipment and several glaciological field campaigns have been carried out (e.g., Figure 1f).

L132-133: Maybe shortly explain how the measured precipitation relates to the mean annual precipitation given for La Laguna earlier in the manuscript (i.e. why 3 times higher, elevation?)

Precipitation at La Laguna DGA was of 63 and 82 mm in 2019-2020 and 2020-2021, respectively, which means that precipitation at TAP was 4 to 5 times higher than at La Laguna. We added these numbers to the manuscript.

L142: somewhere in this paragraph or earlier, suggest to explain winter/summer and the corresponding months

We have added the corresponding months (winter: June-August, summer: December-February) to the first paragraph of section 2 (Study Area).

Section 3.2: it would be good to explain why a model is used when daily SWE maps are also available. Now the reader has to guess based on the Results section. The same explanation is lacking for the SA and SP indices, why are these indices relevant for this study?

The snow evolution model is needed because it provides several variables apart from SWE, being snowmelt and sublimation the most relevant ones for this study. Also, the SWE reconstruction from Cortés and Margulis (2017) ends in 2015.

The SA and SP indices and the SWE reconstruction are used as verification datasets for the model results.

Apart from the Introduction, this information has been included in a more explicit way at the beginning of the corresponding sections (sections 3 and 4).

3.3 Not a great name for a section and confusing as it still describes another snow product. It would be useful to describe what different information can be obtained from the SA and SP indices and the SCA

We have decided to remove this section. We have moved the SCA data to the previous section (Snow products) and the SRTM DEM is now presented in the SnowModel section. The SA and SP summarize information over several months, whereas the SCA values are used as instantaneous information.

L239: why 5 mm and in table 3 1mm?

In the original manuscript, Table 3 was showing only one of the values of surface roughness that are used in this study, whereas Table 4 was showing the three values that are used to build the ensemble results. We note that the albedo fit was made using the middle value (5 mm). To improve clarity, in the revised version we have merged Table 3 and 4 (as suggested by the other reviewer).

4.2: is there any routing of the snowmelt runoff?

No, in our study there is no routing of the snowmelt runoff.

Table 3: why is the precipitation lapse rate 0?

We have included this information in the revised version:

"Although there is an annual precipitation lapse rate from the lowlands of the Coquimbo Region up to La Laguna DGA station (3160 m a.s.l.), we used a value of zero because we do not have data to support a precipitation lapse rate above that elevation, particularly within the relatively small area of the Corrales catchment. In general, snow distribution at high-elevation catchments is governed mostly by wind transport (e.g. Lehning et al., 2011)."

Table 4: how were these values determined? And is the precipitation correction coming on top of the 30% increase (bias correction) in the precipitation measurements?

Yes, the precipitation factor is on top of the 30% increase estimated due to undercatch (Macdonald and Pomeroy, 2007). As the 30% value was chosen based on previous studies, we selected precipitation factors that can be interpreted as an undercatch uncertainty range. The chosen surface roughness lengths vary within the typical ranges given in literature for snow and ice surfaces (e.g. Brock et al., 2006; Fitzpatrick et al., 2019). The slope and curvature weights for wind distribution were chosen to explore the sensitivity of snow ablation to these parameters in the Andes mountains. This information has been added to the manuscript in Section 4.3.

Figure 3: some suggestions:

Add row names with Snow absence and snow persistence, and add in caption the period for which this was calculated (months, years)

Could one color bar be used for all graphs? Why are color bars for c and d smaller? Could the bars be made such that the colors are more intuitive, i.e. the same color to indicate snow accumulation "hotspots" (high SP values and low SA values?)

Add axis labels on the axes and not at the top

What are the white areas in a?

What are the numbers in b and e? elevation?

Add explanation about the units of each graph in the caption, i.e. indicate that it is a fraction of time in a,b, d and e and percentage of space in c and f.

Thanks for your suggestions. This figure has been merged with Figure 4a-b-c and we have performed all the suggested changes in the figure and the caption. Please see the new figure and caption below.

[Figure]

Figure 3: SA, SP and SWEmax in the Corrales basin. (a-d-g) Maps, (b-e-h) Polar plots, (c-f-i) Histograms. In the polar plots (b-e-h) the angles represent the orientation, and the elevation is represented by the inverse radial distance. While SA and SP refer to the percentage of time when snow is absent or present in the accumulation (April-September) and melt (October-March) periods, respectively, the histogram provides information about the spatial distribution of the variables across the catchment. Glacier outlines and contours are shown in (a), (d) and (g). Blank areas in (a) represent sites where the SA index cannot be calculated due to an insufficient number of cloud-free images in April-September.

L269: "interannual median" – per pixel or for the catchment?
Per pixel. We have included this information.

Figure 4, similar issues as with Figure 3. In particular, please provide units in the graphs.
Figure 4a-b-c has been merged with Figure 3. Figure 4d-e-f (coefficient of variation) has been moved to the Supplement. We have corrected the problems found by the reviewer.

L290: where does "also" refer to?
It was superfluous, we have removed it.

Figure 6 – it would be more logic to have a maximum y-axis of 100% in figure a.
Agreed. We only added it to accommodate the legend, but we now placed it above the chart.

L307-309: Why was the comparison not based on simulations at the same time as the satellite data? This would be a more fair comparison and indicate if the model is under- or overestimating snow persistence. The same comment for the SWE estimates in the next lines.
To make a more fair comparison, we have calculated the SA and SP indices for winter and summer, respectively, using the same dates as the satellite data. As a result, the observed and satellite-based absolute values are now much similar (see below). Unfortunately, in the case of SWE we can't do the same because the SWE reconstruction from Cortés and Margulis (2017) ends in 2015. The comparison is only to assess whether the relative distribution of SWE across the catchment is the same for the SnowModel simulations and the SWE reconstruction product. We now explain this more explicitly in the main document.

[Figure]

Figure 7: What is meant with "to help comparison" in the caption?
Please note that Figure 7 has been moved to the Supplement of the revised version. We did this to keep the focus of the main document on the validation of snow variables. "To help comparison" was meant to explain why the first reading was placed at the simulated value on that date and not at zero. Alternatively, we could start the plot at the date of the first reading, but the comparison will look the same, but with an offset. The main message is that the difference between readings is well simulated by the model.

Why is there only one stake for 2020 at the beginning of the season?
We drilled a couple of more stakes at the beginning of the 2020 season (December 2019), but they had already collapsed when we visited them in January 2020. In that same visit we drilled two additional stakes.
Why is the first value not the measured value but the simulated value.
Please see our previous response. This was just meant for visual purposes, but the main message is that the difference between readings is well simulated.

Figure 8: some suggestions:
Add more x-axis labels
Indicate somewhere that these are stacked bars
We have performed the suggested changes. Please note that this is now Figure 6.

During summer snowfall events (Jan 2020) it looks like the ice melt flux is largest – why is that?
Although there was a snowfall on January 25-27 2020, the month was dominated by clear-sky conditions that led to large ice melt rates. We have included this in the text.

L361: "where snowmelt is more important than sublimation" – if values are above 50% than this is nowhere the case?
Agreed. We have changed this to "i.e. where snowmelt is more important than sublimation" by "where snowmelt runoff is highest".

L365: 4.3 mm3 a-1 – to give a hydrological meaning to the number, wouldn't it make more sense to give it in the same units as P, i.e. mm per catchment?
Agreed. We have changed the units to mm a$^{-1}$, but we have kept the Mm3 value in parenthesis as these units are useful to make comparisons with the capacity of La Laguna reservoir.

Figure 9 – Please introduce some consistency in the color bars and ranges. Also the headers could be improved which are now sometimes over two lines and sometimes centered but not always.
Thanks for your suggestions. We have improved the visualization of this figure.

[Figure]

L368 "total ablation" and L365 "total ablation" – please reformulate as they refer to something different. This also applies to the conclusion, point 1 which is now a bit ambiguous
We have reformulated the text. In the revised version the term "total ablation" refers only to the sum of melt and sublimation.

L370-375 how are the location of the snowmelt hotspots determined? Is there a clear ranking of which cells are included and which not? The line in Figure 10 c looks rather linear, at least between 10 and 30% of the area?

We first rank the grid cells based on their snowmelt runoff and then we define as snowmelt hotspots the first grid cells of the ranking that produce 50% of the total snowmelt runoff. We have included this information in the revised text.

Figure 10: please remove the text from the figure C
Agreed. The text has been removed.

L387: Do the SP and SA values refer to the observations or the simulations?
The values refer to the simulations. We have included this information in the revised caption.

L392 "consequently" – Please explain this last sentence in more detail
This was not clear. We have reworded it to "Sublimation ratio is largely dependent on z0 (R2=0.85), which controls the magnitude of turbulent latent heat fluxes."

Figure 11: why are cumulative distributions used here? Please also add in the caption which data is in the graphs. For example for the maximum snow depth, is it the maximum depth over the whole simulation period?
We use cumulative distributions because we think that they allow a fast identification of percentages. For example, in the new Figure 6a (below), "80% of the snowmelt hotspots are located below 4600 m a.s.l."

[Figure]

The role of Figure 12 in the study does not become clear from the text.
In the revised version we have improved the text to better justify the inclusion of this figure. Most importantly, these results go in the same direction as the rest of the variables, i.e. snowmelt hotspots are located on sites where snow erosion is lowest, whereas the opposite slopes present large values of snow surface sublimation, erosion and blowing snow sublimation.

L453: "similar AWS forcing" – what is meant here, as the lines before just explain that there was a different availability of AWS data. "Similarly" – similar to whom?
We have removed and reworded: "In fact, some differences between our simulations and those of Réveillet et al. (2020), who also used AWS forcing in the same catchment, might be caused

by gaps in the PAN records during their study period. For example, we obtain much larger sublimation ratios (~85% versus ~35%). "

L477: "In this direction…..this type of environment" – it comes across as if this sentence does not fit the text
We have removed it.

L487: "We here show that …" – please describe more explicitly what is the case here in this study, also referring to figures.
The revised Figure 9a now shows more explicitly what we meant. If sublimation rates are low, the ablation of the end-of-winter snow cover is largely dominated by melt and total seasonal snowmelt are expected to show almost the same spatial distribution as snow accumulation. The new text in the Discussion addresses these findings in more detail.

L491: "where large part of snowmelt is generated" – repeat where this is and if this is general for semiarid Andes
We referred to the snowmelt hotspots. We have reworded these sentences to make them clearer.

L507: can ice also sublimate?
Yes, but it seems that the large surface melt rates caused by the low ice albedo dominate over ice sublimation. Please note that SnowModel does consider ice sublimation and this was much lower than ice melt once the snow disappeared from the surface. We have added a short sentence about this in parenthesis: "ice sublimation calculated by SnowModel was much lower than ice melt".

In section 6.2, I was expecting a discussion about the "glacier hotspots" too, as they turned out to provide even more melt, but have an even smaller area. It is shortly mentioned, but what is the hydrological implication of glacier hotspots versus snowmelt hotspots?
Thanks for this interesting suggestion. Indeed, Tapado Glacier can be considered a hotspot as well. We have incorporated this point to the discussion based on the revised Figure 6. The role of ice melt during dry years is indeed very important in this region.

**References**
Brock, B. W., Willis, I. C. and Sharp, M. J.: Measurement and parameterization of aerodynamic roughness length variations at Haut Glacier d'Arolla, Switzerland, J. Glaciol., 52(177), 281–297, doi:10.3189/172756506781828746, 2006.
Cortés, G. and Margulis, S.: Impacts of El Niño and La Niña on interannual snow accumulation in the Andes: Results from a high-resolution 31 year reanalysis, Geophys. Res. Lett., 44(13), 6859–6867, doi:10.1002/2017GL073826, 2017.
DeBeer, C. M. and Pomeroy, J. W.: Influence of snowpack and melt energy heterogeneity on snow cover depletion and snowmelt runoff simulation in a cold mountain environment, J. Hydrol., 553, 199–213, doi:10.1016/j.jhydrol.2017.07.051, 2017.
Fitzpatrick, N., Radić, V. and Menounos, B.: A multi-season investigation of glacier surface roughness lengths through in situ and remote observation, Cryosphere, 13(3), 1051–1071, doi:10.5194/tc-13-1051-2019, 2019.

Freudiger, D., Kohn, I., Seibert, J., Stahl, K. and Weiler, M.: Snow redistribution for the hydrological modeling of alpine catchments, Wiley Interdiscip. Rev. Water, 4(October), e1232, doi:10.1002/wat2.1232, 2017.

Lehning, M., Grünewald, T. and Schirmer, M.: Mountain snow distribution governed by an altitudinal gradient and terrain roughness, Geophys. Res. Lett., 38(L19504), 1–5, doi:doi:10.1029/2011GL048927, 2011.

Macdonald, J. and Pomeroy, J.: Gauge Undercatch of Two Common Snowfall Gauges in a Prairie Environment, Proc. 64th East. Snow Conf. St. John's, Canada., (1974), 119–126, 2007.

Mendoza, P. A., Shaw, T. E., McPhee, J., Musselman, K. N., Revuelto, J. and MacDonell, S.: Spatial Distribution and Scaling Properties of Lidar-Derived Snow Depth in the Extratropical Andes, Water Resour. Res., 56(12), 1–23, doi:10.1029/2020WR028480, 2020.

Mott, R., Schirmer, M., Bavay, M., Grünewald, T. and Lehning, M.: Understanding snow-transport processes shaping the mountain snow-cover, Cryosph., 4(4), 545–559, doi:10.5194/tc-4-545-2010, 2010.

Pohl, S., Marsh, P. and Liston, G. E.: Spatial-temporal variability in turbulent fluxes during spring snowmelt, Arctic, Antarct. Alp. Res., 38(1), 136–146, doi:10.1657/1523-0430(2006)038[0136:SVITFD]2.0.CO;2, 2006.

Réveillet, M., MacDonell, S., Gascoin, S., Kinnard, C., Lhermitte, S. and Schaffer, N.: Impact of forcing on sublimation simulations for a high mountain catchment in the semiarid Andes, Cryosph., 14(1), 147–163, doi:10.5194/tc-14-147-2020, 2020.

---

## Author Response (AR2)

**Reviewer 1**

Dear authors and editor,

I appreciate the effort of the authors in addressing the issues and questions raised in the first review round. The responses are clear and the revisions have significantly improved the manuscript. I only have a couple of minor comments, which should be addressed before acceptance.

We thank the reviewer for reviewing again our article and their new comments. Please see our responses below. In addition to your suggestions, we also noticed that the numbers for total annual precipitation given in the text (Section 3.1: Field data) were incorrect. In the revised version we have corrected these numbers (they changed from 309 and 330 mm to 277 and 335 mm for water years 2019-2020 and 2020-2021, respectively). This was a textual error and did not affect the rest of the manuscript.

L144: What is La Laguna DGA? Is this at La Laguna reservoir? This full term has not been introduced previously.

Thanks for noticing this. La Laguna DGA is a meteorological station next to the La Laguna reservoir that is maintained by the Chilean Water Directorate (Dirección General de Aguas, DGA). We briefly introduced the station in the previous version, but without using the full term "La Laguna DGA". We have introduced the term correctly in the latest version (Section 2: Study Area).

L151-153: The gap at TAP between December 2020 and January 2021 is the same as the large data gap in summer 2021, right? Perhaps, you could use the same phrasing.

Yes, it is the same gap. To avoid confusion, we have decided to remove the second comment about the missing data.

Fig 2: The dark blue and black colored lines are quite hard to separate from each other (especially in the legend). Would it be possible to redo the figure with changing either one of these colors into another more distinct color?

Yes, we have changed the dark blue color to a lighter blue that is easier to distinguish from the black lines.

[Figure]

**New Figure 2**

L237-238: The snow surface sublimation that you extract from the turbulent latent heat flux, is that a direct output term of SnowModel or do you have to do some additional computations? If the latter, how did you compute it?

No there was no additional calculation for snow surface sublimation. SnowModel produces this variable directly. We have deleted the second parenthesis in this sentence. The new sentence reads as follows: "In our study, we analyze results for snowmelt runoff (snowmelt leaving the snowpack) and snow surface sublimation".

L256 & 4.3: Did the windspeed input also vary with the variations in z0? Or did you use only z0 = 5 mm for the wind speed calculation and keep this constant for all ensemble runs?
We did not recalculate the 2-m height wind speed for each ensemble run. We have now added a small comment about this in the new version of the article.

Fig 3: I think Figure d and f show contrasting results. I would expect f to be exactly inversed, based on figure d. Or am I misinterpreting them? Perhaps it would be most intuitive if the colorscale of d is also inversed (similar to the previous version, but then still from 0 to 1).
Thanks for noting this. The reviewer is correct. The colorbar of Figure 3d is inverted. We have fixed this problem in the latest version. Please see the figure below.

[Figure]

L384: "at the border with Argentina" might not be clear to the reader.

We have deleted that sentence.

Fig 8b&c: These figures are a bit hard to understand due to the selected colorscales, and make it difficult for the reader to observe themselves what you state in the preceding text in L390-400. We agree with the reviewer that the patterns described in the text were difficult to follow in the figure. We think that it was difficult to understand because some descriptions were not clear and therefore we have modified the text slightly to make it more consistent with the figure. We prefer to keep the colormap as it was chosen (red colors and starting from 0) to distinguish sublimation processes that decrease the water availability of the catchment with red colors from snowmelt runoff with blue or green colors. Also, we chose the red colormap starting from 0 to highlight the overall high values of both, snow surface sublimation and sublimation fraction. Please see the revised text and the figure below:

"Once snow starts to metamorphose in response to internal exchanges of energy and vapor, wind transport is reduced, and the snowpack is more favorable to surface sublimation and snowmelt runoff. We find that surface sublimation was the biggest loss of snow mass, showing large values at the north-western edges of the catchment (Figure 8b). Snowmelt runoff shows a heterogeneous distribution with large values at wind-protected valleys in the north-west section of the catchment and very low values to the east (Figure 8a). On average, the elevation band with the largest values of snowmelt runoff was that between 4000 and 4500 m a.s.l. with a SE aspect (Figure 8d). The sublimation fraction is above 60% across the entire domain, and above 80% at the high-elevation north-western edge of the catchment, where surface sublimation is very large (Figure 8b), and to the eastern areas of the catchment, where snowmelt runoff is almost zero (Figure 8c). Glaciers occur in sites dominated by snow surface sublimation with a sublimation fraction larger than 80% (Figure 8c). However, in terms of runoff volume, we find that ice melt corresponds to 60% of the runoff contribution from the cryosphere (snowmelt and ice melt), which is equivalent to 55 mm a-1 (or 4.3 Mm3 a−1, about 10% of the maximum capacity of La Laguna reservoir)."

[Figure]

**Reviewer 2**

I think the manuscript has greatly improved. The structure of the results and discussion session reads much better now and the figures have also been cleaned up and are therefore more clear. We thank the reviewer for taking the time to review our article again and their positive comments. Please see our responses below. In addition to your suggestions, we also noticed that the numbers for total annual precipitation given in the text (Section 3.1: Field data) were incorrect. In the revised version we have corrected these numbers (they changed from 309 and 330 mm to 277 and 335 mm for water years 2019-2020 and 2020-2021, respectively). This was a textual error and did not affect the rest of the manuscript.

I have a few more small comments:
- In the abstract, the fifth sentences reads a bit difficult, "permit,xxxx, limited removal" – consider rewording
We have simplified this sentence to: "In this study, we suggest that most of the snowmelt runoff originates from specific areas with topographic and meteorological features that allow large snow accumulation and limited mass removal due to high sublimation fraction".

- L18 (Track change version) – winter snow and summer snow? Instead of snow winter absence?
Thanks for this suggestion. We have changed "snow winter absence" and "snow summer persistence" by "winter snow absence" and "summer snow persistence", respectively throughout the entire document.

- L25 – "rock glaciers" – although there is some explanation how snowmelt hotspots could be related to rock glaciers in the discussion, I was a bit confused about it for this manuscript, as the snowmelt hotspots seem not to be in areas where rock glaciers occur. Could this also be left out in the abstract and conclusion for this paper as this has not been explicitly shown?
We thank the reviewer for this comment as it gives us the opportunity to show an additional figure that is not included in the article. We found that there are several rock glaciers in the areas identified as snowmelt hotspots. In the figure below we show the outlines of rock glaciers that have been previously identified in this region (CEAZA, 2020). Clean glaciers in the catchment are shown in white. This confirms our ideas of snowmelt playing an important role in the internal water paths of rock glaciers. We added this figure to the Supplementary Material (Figure S8).

[Figure]

- L80 – "Runoff generation processes" – what is meant here? Still no hydrological model is used and no streamflow is simulated. Would "snowmelt processes" not be a better word?
We have reworded this sentence to: "Given that snowmelt explains 85% of streamflow variability in semiarid catchments (Masiokas et al., 2006) and is a useful predictor of streamflow (Sproles et al., 2016), it is of vital importance to properly understand snow processes and quantify snowmelt volumes in their full complexity and spatial variability."

- L81-L86 – here I read two times a very similar sentence. I think it could be shortened into one sentence
We have merged both sentences to: "In this work, we hypothesize that the meteorological and topographical conditions of the semiarid Andes result in large areas where snow surface sublimation losses dominate over snowmelt, thus delimiting relatively small areas from where most of the snowmelt runoff is generated, and further increasing the typically large spatial variability of snowmelt in mountain terrain."

- L151 – I was a bit confused to read here that precipitation amounts were increased by 30%, but later on I read that the precipitation correction factor was set to 0.7. Are these two options not cancelling each other out?
Yes, that is correct, both factors cancel each other in the ensemble runs with a precipitation factor of 0.7. This is because the factors used in the ensemble runs were chosen as an uncertainty range for snow undercatch, i.e. from 0 to 60%.

- L178 – Could you add "at a monthly resolution"?
Done.

- L206 – Maybe change "finally" into "third", to be consistent

Done.

- In Section 5.1 where does the 0.6 Threshold come from?
This number was used only to help in the description of the figure. In the revised version, we have only kept the first use of the number and replaced the rest by other numbers that are more intuitive, such as SA=0.5 (half the wintertime).

- Figure 3 – I still have a bit trouble with the color bars for SA and SP. For SA, a high number (yellow) means that most of the time snow is absent. That same color means for SP that snow is present (in almost the whole basin?). Could one of the bars not be flipped?
Thanks for noting this. Indeed, the colorbar in Figure 3d was flipped. We have fixed this in the revised version.

[Figure]

- L432 – "Rainfall was much lower than snowfall", but later it is stated that "snowmelt and icemelt played a secondary role at the annual level". How does this work?
Yes, this was not fully clear. The second statement is in comparison with the other mass losses, mainly surface sublimation. We have now made this more explicit in the full sentence: "Snow

surface sublimation was the process that removed most of the snow mass and dominated ablation during winter and spring. In comparison, snowmelt runoff and ice melt played a secondary role at the annual level, but their relative importance increased in spring and summer."

- I think Figure 6 is very interesting. Would it be possible to say something about the flow during winter? Do these amount match with monthly precipitation/rainfall inputs for the station further downstream? Maybe it could also be mentioned that almost all of the flow into the reservoir in summer comes from the upstream catchment

Thanks for this comment. This is an interesting subject. We have isotopic analyses (unfortunately not yet published) showing that the winter base flow is sustained by groundwater that originates from previous melt seasons ("old water"). This base flow is occasionally interrupted by snowmelt originating from elevations below 4000 m.a.s.l. Since the 0°C isotherm during precipitation events is typically below 3000 m a.s.l., we do not expect large contributions from rainfall.
We prefer to not to comment on the second suggestion as the contribution from lateral sub-catchments may be important occasionally depending on the amount of accumulated snow.

- L526-528: "For this xxxx snowmelt runoff" – this is already in the methods
Removed.

- L533 "Interestingly" – I found this confusing since it was just explained above that almost all snow sublimates at the glacier surface
Removed.

- Figure 10 – Indicate the threshold of 50% also in the figure, otherwise it is not understandable without reading the text
Done

[Figure]

- L585 – ExplainS
Corrected.

- L593 "General spatial trends" – I wondered if it would be possible to provide somewhere a map where model simulations of SWE/SA/SP and observations could be compared?.
Yes. We have added a new figure to the SI with the maps (also here below). The maps show the same conclusions derived from Figure 5, i.e. there is a general agreement in the spatial trends, but the exact values do not match completely, with the differences discussed in the main text. Small-scale variability is also difficult to reproduce.